# Pragmatic Language Impairment: A Scientometric Review

Ahmed Alduais [1,*], Hind Alfadda [2,*], Silvia Allegretta [3] and Tamara Trivkovic [4]

1    Department of Human Sciences, University of Verona, 37134 Verona, Italy
2    Department of Curriculum and Instruction, King Saud University, Riyadh 11362, Saudi Arabia
3    Department of General Psychology, University of Padua, 35128 Padua, Italy
4    Department of Speech and Language Pathology, College of Social Work, 11000 Belgrade, Serbia
*    Correspondence: ibnalduais@gmail.com or ahmedmohammedsalehalduais02@univr.it (A.A.);
    halfadda@ksu.edu.sa (H.A.)

**Abstract:** Pragmatic language impairment (PLI) is a complex and wide-ranging condition affecting numerous individuals worldwide, yet its exact prevalence and scope remain uncertain due to its interconnections with other conditions and symptoms, such as neurodevelopmental disorders, learning disabilities, developmental dysphasia, and aphasia. This study presents a comprehensive review of PLI, tracing its historical, current, and future trajectories through the lens of both bibliometric and scientometric indicators. The study analysed a substantial corpus of 3852 documents related to PLI, including sources from Scopus, Web of Science, and Lens, spanning the period from 1977 to 2022. This investigation utilised advanced software tools such as CiteSpace 5.8.R3 and VOSviewer 1.6.18 to detect patterns, connections, and bursts in scholarly works related to PLI. Key findings of this review include the identification of major clusters in the PLI literature, which include social communication disorder, traumatic brain injury, autism spectrum disorder, and inferential meaning. These clusters represent significant sub-themes within the PLI research body, with repetitive mentions of autism spectrum disorder suggesting its critical overlap with PLI. Other noteworthy clusters included Asperger's syndrome, behavioural problems, belief reports, and diagnostic observation schedule scores, all of which contribute to the nuanced understanding of PLI. The study provides a comprehensive overview of PLI development, drawing on theoretical, historical, and empirical evidence.

**Keywords:** pragmatic language impairment; social pragmatic communication disorder; semantic-pragmatic disorder; aphasia; scientometric review

## 1. Introduction

Pragmatic Language Impairment (PLI) is an intriguing field of study that has seen significant growth and evolution over the years. This paper embarks on an extensive exploration of the research landscape surrounding PLI. The paper's introduction is structured around six key themes that provide a comprehensive background for the scientometric review. "The Rise of Pragmatic Language Impairment" explores the historical progression of PLI as a recognised condition, while "The Scope of Pragmatic Language Impairment" delineates the breadth and depth of this intricate disorder. The complex relationship between PLI and Autism Spectrum Disorder is unravelled in "Pragmatic Language Impairment and Autism Spectrum Disorder". The section on "Diagnostic Instruments for Pragmatic Language Impairment" introduces the tools that have been instrumental in diagnosing PLI, and "Scientific Contributions for Pragmatic Language Impairment" acknowledges the seminal works that have shaped the field. Finally, "Purpose of the Present Study" articulates the driving aim of our review: to map the research trajectory of PLI, identify key trends, and highlight future directions for exploration. This robust introduction serves as a springboard for the comprehensive scientometric analysis that follows, offering a panoramic view of the PLI research landscape.

### 1.1. The Rise of Pragmatic Language Impairment

Pragmatic Language Impairment (PLI) is a disorder that has been referred to by a myriad of terms and has been defined in various ways throughout its history, demonstrating the evolution of our understanding of this condition [1–5]. Currently, it is officially recognised as Social (*Pragmatic*) Communication Disorder (S*PCD*) in the fifth edition of the Diagnostic and Statistical Manual of Mental Disorders (DSM-5) [6]. The existence of this disorder was first identified by Rapin and Allen [7], who classified it under the taxonomy of developmental speech disorders and coined the term Semantic Pragmatic Syndrome. They emphasised that this condition typically manifests in individuals who, despite having largely intact structural language abilities, struggle with semantic and pragmatic aspects of communication. This led to the creation of a new term, which will be used in the future [8,9]—Pragmatic Language Impairment (PLI) [10]. Other researchers are of the opinion that PLI cannot be equated with language disorders because people with this disorder do not have a problem with language structures but with its use [11].

Previous research demonstrates that the same symptoms occur in PLI as in autism spectrum disorders (ASDs) [12]. This marked PLI's future as a disorder that is often associated with autism spectrum disorders. On the other hand, other researchers claim that there is evidence to the contrary. Among these claims is that children with PLI do not necessarily show a triad of impairments (communication, social interaction, and interests) which is noticeable in people with autism (Bishop [13]). Further, it is suggested that individuals with PLI fall along a continuum between individuals with Specific Language Impairment (SLI) and individuals with ASD [14]. Due to a lack of clarity about terminology and diagnostic criteria for PLI, there is still debate as to whether it is actually a language disorder or an autism spectrum disorder and whether PLI should be a separate diagnostic entity.

### 1.2. The Scope of Pragmatic Language Impairment

Pragmatics was primarily studied within the disciplines of linguistics, anthropology, and literature. As the scientific field developed and the need to understand the multitude of disorders, their causes and deviations from the norm arose, experts in other areas began examining pragmatics. Philosophy, sociolinguistics, psychology, special education, rehabilitation, and speech–language pathology explored pragmatics based on their respective foci. For example, speech–language pathologists investigated pragmatics in relation to communication; sociologists considered pragmatics with respect to social roles; anthropologists studied pragmatics in terms of culture; and psychologists explored pragmatics regarding developmental disorders. Skills associated with pragmatic competence are diverse, encompassing turn-taking, nonverbal behaviours, topic and theme management, cohesive ties, and presuppositions [15].

As mentioned earlier, this disorder is currently known as S*PCD*. The main symptoms include impaired social use of verbal and nonverbal communication [16]. Impaired social use of verbal and nonverbal communication refers to difficulties in using language and gestures in social interactions. It is not about the ability to speak, understand, or recognise gestures, but rather about using these tools appropriately in various social situations. For example, verbal communication involves not just speaking, but also using the right tone, volume, and pace suitable for the situation. A person with an impairment might speak too loudly in a quiet setting or use a formal tone in a casual conversation. Nonverbal communication includes gestures, facial expressions, and body language. An individual with this impairment might not make eye contact when speaking or might not understand the meaning behind certain facial expressions or gestures. They might stand too close to someone, not respecting personal space, or fail to use appropriate gestures that typically accompany speech, leading to awkward or misunderstood social interactions [1–5].

A previous version of this definition was given by Rapin and Allen [7] and Bishop and Rosenbloom [8]. Rapin and Allen introduced the following symptoms when defining the term Semantic Pragmatic Syndrome: (1) comprehension deficits of connected discourse, (2) verboseness, (3) word-finding deficits as evidenced by circumlocutions, semantic para-

phasias, and lack of semantic specificity, (4) stereotyped conversational responses, (5) literal interpretations, (6) responses to one or two words, (7) impairment in the ability to take turns and to maintain a topic in discourse, and (8) unimpaired syntax and articulation [17].

Children with Pragmatic Language Impairment have a problem with the use of language in social situations, understanding of the context and discourse of language [10,18], and difficulties following rules for conversation and storytelling, such as taking turns in conversation (American Psychiatric Association, 2013) [6]. The main reason for this is that they share very little information, so it is difficult for the listener to understand what happened because there are not enough vital descriptions from which to compose a story [6]. According to the American Psychiatric Association, these children have problems in the ability to change communication to match the context or the needs of the listener (whether it is a child, a job interview, or a woman working in a store) [6]. Contrary to previous assertions—recently debunked—that posited children with pragmatic language impairment largely remain undiagnosed until school age [19], contemporary research has convincingly refuted this claim. The current consensus attributes the delayed diagnosis to these children's capacity to exhibit language fluency and utilise conventional syntax [20–22]. Such abilities can effectively obscure the presence of the impairment, thereby leading to a delay in the recognition and diagnosis of their condition.

### 1.3. Pragmatic Language Impairment and Autism Spectrum Disorder

Many clinicians and researchers have suggested that PLI should be considered an autism spectrum disorder (ASD). PLI shows considerable overlap with autism spectrum disorders. Without a doubt, pragmatic language problems are one of the key and striking problems in individuals with autism. According to the definition by the American Psychiatric Association [23], one of the symptoms of people with autism is a lack of understanding of the context of conversation and stereotypical and repetitive communication. This is supported by numerous studies showing the overlap of symptoms in people with PLI and people with ASD [12]. This overlap is also confirmed several studies including Botting and Conti Ramsden [24], who found that roughly half of their group of children with PLI qualify for ASD diagnosis (based on DSM-V [6]). This certainly calls into question whether all children who have a pragmatic language disorder also have an autism spectrum disorder.

What is especially important is that when ASD is diagnosed, there must be other symptoms such as the inability to achieve social interaction and limited interests, as well as stereotypical actions. Most children with PLI do not have any of these symptoms. Bishop [10] and Bishop and Norbury [14] proved that children with PLI do not necessarily show the triad of impairments (communication, social interaction, and interests) that has been reported in individuals with autism. However, most children with PLI do show evidence of stereotyped language, a symptom that has been reclassified as a repetitive behaviour in DSM-5 [6]. Many of the children with PLI in the samples used by Bishop and Bishop and Norbury may be considered to have an ASD, according to the new criteria-the required number of symptoms within this domain is two [25].

Recent research shows that the line between PLI and ASD is very thin. Reisinger, Cornish, and Fombonne [26] found similar levels of restricted interests and repetitive behaviour in children with PLI and children with ASD. Certainly, the ASD group displayed more severe social and communication deficits. The problem in diagnosing PLI and ASD can be significant when people with mild forms of ASD are diagnosed with PLI and do not receive adequate treatment in relation to the diagnosis. People with PLI are still between two important diagnoses-specific language disorder (SLI) and autism spectrum disorder (ASD) [27].

### 1.4. Diagnostic Instruments for Pragmatic Language Impairment

When problems with the use of pragmatics are suspected during triage observation, it is necessary to carry out some tests. The following tests are used, but are not the only ones, for this purpose: Communication and Symbolic Behaviour Scales [28] for younger children and the Yale Pragmatic Profile [29] for older children. In addition, one of the most popular

standardised tests is the Test of Pragmatic Language—Second Edition (TOPL-2) [30], which examines the broadest aspects of pragmatics.

Communication and Symbolic Behaviour Scales [28] is a standardised tool designed to evaluate the communication and symbolic abilities of children whose functional communication age is between 6 months and 2 years. The Communication and Symbolic Behaviour Scales Developmental Profile can also be used with preschool children whose chronological age is up to 5–6 years, but only if their developmental level of functioning is lower than 24 months. The purpose of this tool is threefold: a screening tool for identifying children at risk for developmental delay or disability; to determine if a child has delays in social communication, expressive speech/language, and symbolic functioning; and for evaluation to document changes in social communication, expressive speech/language, and symbolic functioning over time.

The Early Social Communication Scales for younger children [31] is a structured observation recorded on video that requires between 15 and 25 min to apply. It is designed to provide measures of individual differences in nonverbal communication skills which usually occur in children between 8 and 30 months of age. Tasks include the presentation of object spectacle toys (e.g., a wind-up toy), turn-taking tasks (e.g., ball play), social interaction (e.g., tickling), gaze-following tasks, and opportunities to respond to an invitation to play. This diagnostic tool can also be used with children with typical development within this age or with a child with delayed verbal development whose age is in this age range.

The Yale Pragmatic Profile [29] is a diagnostic tool that measures pragmatic language via a semi-structured conversational task in verbal children ages 9–17. It contains a series of predetermined probes to collect information on a variety of conversational speech acts. Within this 30–40-min conversation, the examiner inserts 23 pragmatic probes within five conversational domains (discourse management, communicative function, conversational repair, presupposition, register variation).

A new version of the TOPL-2 [30] involves four aspects that are observed—researching pragmatic language skills, identifying individuals with pragmatic language deficits, determining individual strengths and weaknesses and documenting an individual's progress. This test was originally designed for use by speech–language pathologists/logopedist, but it is no longer used only by these experts, but also by all associates such as psychologists, counsellors, clinical psychologists, and specialists in the field of special education and rehabilitation. These varied team members now use the TOPL-2 as part of a full individual evaluation and program planning, which were part of TOPL-2 development and norming.

*1.5. Scientific Contributions for Pragmatic Language Impairment*

When we talk about high-quality journals Q1 through Q4, there is not a single journal that deals exclusively with pragmatic language disorders. One of the journals dealing with pragmatics is the *Journal of Pragmatic*, founded in 1977 in the Netherlands, (vol. 193 in progress—May 2022) [32]. This journal has, since its inception, dealt with a wide range of research in pragmatics, including cognitive pragmatics, corpus pragmatics, historical pragmatics, exploratory pragmatics, interpersonal pragmatics, multimodal pragmatics, social pragmatics, hypothetical pragmatics, and related areas. The *Journal of Pragmatics* empowers work that employs verified dialect information to investigate the relationship between pragmatics and neighbouring areas of inquiry such as semantics, conversation analysis and ethnomethodology, interactional linguistics, discourse in communication, linguistic anthropology, media studies, psychology, sociology, and the philosophy of language [32].

Additionally, another journal that deals with, among other things, pragmatics in the form of communication is the journal *Language and Speech*, founded in 1958 in the United States of America (Vol. 65, Issue 1-March 2022, pp. 3–260). *Language and Speech* is a peer-reviewed journal that provides an international gathering of information for communication among analysts within the disciplines that contribute to our understanding of human production, perception, processing, learning, use, and speech and language disorders. Corpus-based, experimental, and observational research within the domain of

linguistic, psychological, or computational models are particularly welcome in this journal, as the editors note [33].

When we talk about associations and support groups, unfortunately, there is still no official information that they exist. What exists and contributes to the understanding and research of pragmatics is "The International Pragmatics Association" (IPrA). This is an international scientific organisation devoted to the study of language use. Established in 1986, in Antwerp, Belgium, it now has approximately 1500 members in over 70 countries over the world. It represents the field of pragmatics, i.e., the science of language use, in its widest interdisciplinary sense as a functional perspective on language and communication–cultural, cognitive ad social [34].

### 1.6. Purpose of the Present Study

There has been a great deal of research conducted on PLI without reaching a consensus. In response to the complexity of PLI, several researchers have attempted to review certain aspects of it. As an example, Adams asserted that the difficulty of developing an accurate assessment tool for PLI is due to its heterogeneous nature (i.e., incorporating linguistics, cognitive, social, cultural aspects). Toward the end of her review, she recommended the proliferation of assessment instrumentation [35]. A second review examined the divergence of theories in examining PLI, including: weak central coherence, social inference, and executive dysfunction [36]. Additionally, there are a number of existing instruments that are used to diagnose and assess persons with PLI that are divergent and result in varying outcomes [37]. Due to the fact that PLI is central to (social) communication for individuals of all ages, it is not only limited to children and adolescents with developmental disorders but also to patients suffering from aphasia, namely damage to the right hemisphere [38].

Researchers disagree on the conceptualisation of PLI, resulting in the use of several concepts, the most popular being [39] social (pragmatic) communication disorder [25,40,41], semantic–pragmatic disorder [7,17,19], and pragmatic disorders [42]. It is important to note that although a great deal of knowledge has been produced on PLI since the 1980s, recent research has sought consensus scoping and better-quality instruments for diagnosing, assessing, and treating PLI [43].

Based on the scope and objectives outlined for this scientometric review, the following research questions can be formulated:

1.  Bibliometric Question: How have production size, geographical contribution, source, and citation trends in the field of Pragmatic Language Impairment evolved over time, and which authors, universities, and publishers have emerged as the most influential contributors to this domain's knowledge base?
2.  Scientometric Question: In the context of PLI research, how have betweenness centrality, burst detection, co-citation, silhouette, sigma, and clusters evolved, and what does this reveal about the consistency, strength, and relatedness of the nodes within this field? Additionally, how does the co-occurrence of author keywords and citation patterns inform us about the development and primary focus areas of this research domain?

## 2. Methods

### 2.1. Research Methods

Scientometrics focuses on the analysis of artefacts, examining not the processes of science and scholarship but rather their outcomes [44]. The objective of scientometrics is to explore the numerical aspects of generating, disseminating, and utilising scientific information to gain a deeper comprehension of the mechanisms governing scientific research as a societal activity [45]. In this realm of research, it remains ambiguous whether the goal is to enhance the quality of published knowledge. Prior research suggested that identifying quality papers in BIS [bibliometrics, infometrics, and scientometrics] is particularly challenging due to the diverse backgrounds of the researchers [46]. Nevertheless, such studies

primarily aim to uncover features of scientometric occurrences and processes in scientific research for more effective management of scientific endeavours [47].

Research of this type is guided by scientometric indicators, which include various elements and types. Elements are specific features being measured, such as publications, citations, references, or potential future research. Meanwhile, type indicators refer to the nature of the measurement, such as quantitative (numerical data) or impact (the influence or effect of the research) [47]. When undertaking such investigations, scholars often employ the concept of "mapping knowledge domains", which involves generating visual representations that illustrate the evolutionary trajectory and structural interconnections of scientific knowledge, using maps as valuable instruments for monitoring the frontiers of science and technology, supporting knowledge administration, and aiding scientific and technological decision-making [48]. Contemporary research in this domain tends to encompass all disciplines, going beyond the confines of medicine, health, and pure science [49]. The current investigation scrutinised the field of PLI as a sub-discipline of pragmatics, merging with (clinical) linguistics, developmental psychology, neuroscience, and so forth.

### *2.2. Measures*

Research grounded in bibliometrics and scientometrics serves to direct the evaluation of knowledge generated within a particular domain or concept (e.g., PLI). Knowledge repositories (e.g., Scopus, WOS, and Lens) typically offer bibliometric indicators [50–53]. Scientometric indicators can often be procured via specialised software tools. For example, in the present investigation, we employed CiteSpace 5.8.R3 [54] and VOSviewer 1.6.18 [55]. Our study made use of the bibliometric and scientometric indicators delineated in Table 1.

**Table 1.** Bibliometric and Scientometric Indicators for Mapping Knowledge Domains of Pragmatic Language Impairment.

| Indicator: 1–8 Bibliometric; 9–16 Scientometric | Operationalisation | Database/Software | | |
|---|---|---|---|---|
| | | **Scopus** | **WOS** | **Lens** |
| Year [1] | Production size by year | √ | √ | √ |
| Country [2] | Top countries publishing in the field | √ | √ | √ |
| University [3] | Top universities, research centres, etc. | √ | √ | √ |
| Source [4] | Top journals, book series, etc. | √ | √ | √ |
| Publisher [5] | Top publishers | X | √ | √ |
| Subject area [6] | Top fields associated with the field | √ | √ | √ |
| Author [7] | Top authors publishing in the field | √ | √ | √ |
| Citation [8] | Top cited documents | √ | √ | √ |
| | | CiteSpace | | VOSviewer |
| Betweenness centrality [9] | A path between nodes which is achieved when located between two nodes [56] | √ | | X |
| Burst detection [10] | Determines the frequency of a certain event in certain period (e.g., the frequent citation of a certain reference during a period of time) [57] | √ | | X |
| Co-citation [11] | When two references are cited by a third reference [58]. CiteSpace provides a document co-citation network for references and an author co-citation network for authors. In VOSviewer, defined thus: "the relatedness of items is determined based on the number of times they are cited together" [55] (p. 5). Units of analysis include cited authors, references, or sources. | √ | | √ |
| Silhouette [12] | Used in cluster analysis to measure consistency of each cluster with its related nodes [54] | √ | | X |
| Sigma [13] | Used to measure strength of a node in terms of betweenness centrality citation burst [54] | √ | | X |

**Table 1.** *Cont.*

| Indicator: 1–8 Bibliometric; 9–16 Scientometric | Operationalisation | Database/Software | | |
|---|---|---|---|---|
| | | **Scopus** | **WOS** | **Lens** |
| Clusters [14] | "We can probably eyeball the visualised network and identify some prominent groupings" [54] (p. 23). | √ | | √ |
| Citation [15] | "The relatedness of items is determined based on the number of times they cite each other" [55] (p. 5). Units of analysis include documents, sources, authors, organisations, or countries. | √ | | √ |
| Keywords [16] | CiteSpace provides co-occurring author keywords and keywords plus. In VOSviewer, defined thus: "the relatedness of items is determined based on the number of documents in which they occur together" [55] (p. 5). Units of analysis include author keywords, all keywords, or keywords plus. | √ | | √ |

### 2.3. Data-Collection and Sample

To retrieve data, three databases were employed: Scopus, WOS, and Lens. Several factors contributed to the selection of these databases. First, Scopus and WOS index publications are based on specific criteria [50–52]. Additionally, Lens houses a more extensive array of data that are not accessible in either Scopus or WOS [53].

Searches were executed on Wednesday, 30 March 2022. There was no requirement to impose language restrictions, provided that titles, abstracts, and keywords were available in English. The results were manually inspected, as only a limited number of outcomes were available in other languages. We took into account articles, review articles, book chapters, books, conference proceedings (full papers), and early access publications of these types. Table 2 delineates the search strings employed for the three databases and other specifications.

**Table 2.** Search Strings for Data on Pragmatic Language Impairment in Scopus, WOS, and Lens Databases.

Scopus

( TITLE-ABS-KEY ( "pragmatic language impairment" ) OR TITLE-ABS-KEY ( "pragmatic impairment" ) OR TITLE-ABS-KEY ( "pragmatic language difficulty" ) OR TITLE-ABS-KEY ( "pragmatic difficulty" ) OR TITLE-ABS-KEY ( "pragmatic language challenges" ) OR TITLE-ABS-KEY ( "pragmatic challenges" ) OR TITLE-ABS-KEY ( "pragmatic language deficit" ) OR TITLE-ABS-KEY ( "pragmatic deficit" ) OR TITLE-ABS-KEY ( "pragmatic language disorder" ) OR TITLE-ABS-KEY ( "pragmatic disorder" ) OR TITLE-ABS-KEY ( "social communication disorder" ) OR TITLE-ABS-KEY ( "pragmatic communication disorder" ) OR TITLE-ABS-KEY ( "social pragmatic communication disorder" ) OR TITLE-ABS-KEY ( "semantic pragmatic disorder" ) OR TITLE-ABS-KEY ( "semantic pragmatic syndrome" ) OR TITLE-ABS-KEY ( "social communicative impairment" ) OR TITLE-ABS-KEY ( "social communicative deficit" ) OR TITLE-ABS-KEY ( "pragmatic language disability" ) OR TITLE-ABS-KEY ( "pragmatic disability" ) OR TITLE-ABS-KEY ( "pragmatic language dysfunction" ) OR TITLE-ABS-KEY ( "pragmatic dysfunction" ) ) AND ( LIMIT-TO ( DOCTYPE , "ar" ) OR LIMIT-TO ( DOCTYPE , "re" ) OR LIMIT-TO ( DOCTYPE , "ch" ) OR LIMIT-TO ( DOCTYPE , "cp" ) OR LIMIT-TO ( DOCTYPE , "bk" ) )
Wednesday, 30 March 2022, 1069 document results, 1977–2022

WOS

"pragmatic language impairment" (Topic) or "pragmatic impairment" (Topic) or "pragmatic language difficulty" (Topic) or "pragmatic difficulty" (Topic) or "pragmatic language challenges" (Topic) or "pragmatic challenges" (Topic) or "pragmatic language deficit" (Topic) or "pragmatic deficit" (Topic) or "pragmatic language disorder" (Topic) or "pragmatic disorder" (Topic) or "social communication disorder" (Topic) or "pragmatic communication disorder" (Topic) or "social pragmatic communication disorder" (Topic) or "semantic pragmatic disorder" (Topic) or "semantic-pragmatic disorder" (Topic) or "atypical social interactions" (Topic) or "social-communicative impairment" (Topic) or "social communicative impairment" (Topic) or "social-communicative deficit" (Topic) or "social communicative deficit" (Topic) or "pragmatic language disability" (Topic) or "pragmatic disability" (Topic) or "pragmatic language dysfunction" (Topic) or "pragmatic dysfunction" (Topic) or "pragmatic language abnormalities" (Topic) or "pragmatic abnormalities" (Topic) or "pragmatic breakdown" (Topic) or "semantic-pragmatic syndrome" (Topic) or "semantic pragmatic syndrome" (Topic) or "pragmatic and conversational deficit" (Topic) or "conversational disability" (Topic) or "verbal and non-verbal communication deficits" (Topic) or "pragmatic aphasia" (Topic) or "pragmatic dysphasia" (Topic) or "pragmatic and social difficulties" (Topic) and Articles or Review Articles or Book Chapters or Early Access or Proceedings Papers (Document Types)
Wednesday, 30 March 2022, 475 results, 1985–2022

**Table 2.** *Cont.*

| Lens |
| --- |
| ( Title: (   AND ( "pragmatic language impairment" AND    ) ) OR ( Abstract: (   AND ( "pragmatic language impairment" AND    ) ) OR Full Text: (   AND ( "pragmatic language impairment" AND    ) ) ) ) OR ( ( Title: (   AND ( "pragmatic impairment" AND    ) ) OR ( Abstract: (   AND ( "pragmatic impairment" AND    ) ) OR Full Text: (   AND ( "pragmatic impairment" AND    ) ) ) ) OR ( ( Title: (   AND ( "pragmatic language difficulty" AND    ) ) OR ( Abstract: (   AND ( "pragmatic language difficulty" AND    ) ) OR Full Text: (   AND ( "pragmatic language difficulty" AND    ) ) ) ) OR ( ( Title: (   AND ( "pragmatic difficulty" AND    ) ) OR ( Abstract: (   AND ( "pragmatic difficulty" AND    ) ) OR Full Text: (   AND ( "pragmatic difficulty" AND    ) ) ) ) OR ( ( Title: (   AND ( "pragmatic language challenges" AND    ) ) OR ( Abstract: (   AND ( "pragmatic language challenges" AND    ) ) OR Full Text: (   AND ( "pragmatic language challenges" AND    ) ) ) ) OR ( ( Title: (   AND ( "pragmatic challenges" AND    ) ) OR ( Abstract: (   AND ( "pragmatic challenges" AND    ) ) OR Full Text: (   AND ( "pragmatic challenges" AND    ) ) ) ) OR ( ( Title: (   AND ( "pragmatic language deficit" AND    ) ) OR ( Abstract: (   AND ( "pragmatic language deficit" AND    ) ) OR Full Text: (   AND ( "pragmatic language deficit" AND    ) ) ) ) OR ( ( Title: (   AND ( "pragmatic deficit" AND    ) ) OR ( Abstract: (   AND ( "pragmatic deficit" AND    ) ) OR Full Text: (   AND ( "pragmatic deficit" AND    ) ) ) ) OR ( ( Title: (   AND ( "pragmatic language disorder" AND    ) ) OR ( Abstract: (   AND ( "pragmatic language disorder" AND    ) ) OR Full Text: (   AND ( "pragmatic language disorder" AND    ) ) ) ) OR ( ( Title: (   AND ( "pragmatic disorder" AND    ) ) OR ( Abstract: (   AND ( "pragmatic disorder" AND    ) ) OR Full Text: (   AND ( "pragmatic disorder" AND    ) ) ) ) OR ( ( Title: (   AND ( "social communication disorder" AND    ) ) OR ( Abstract: (   AND ( "social communication disorder" AND    ) ) OR Full Text: (   AND ( "social communication disorder" AND    ) ) ) ) OR ( ( Title: (   AND ( "pragmatic communication disorder" AND    ) ) OR ( Abstract: (   AND ( "pragmatic communication disorder" AND    ) ) OR Full Text: (   AND ( "pragmatic communication disorder" AND    ) ) ) ) OR ( ( Title: (   AND ( "social pragmatic communication disorder" AND    ) ) OR ( Abstract: (   AND ( "social pragmatic communication disorder" AND    ) ) OR Full Text: (   AND ( "social pragmatic communication disorder" AND    ) ) ) ) OR ( ( Title: (   AND ( "semantic pragmatic disorder" AND    ) ) OR ( Abstract: (   AND ( "semantic pragmatic disorder" AND    ) ) OR Full Text: (   AND ( "semantic pragmatic disorder" AND    ) ) ) ) OR ( ( Title: (   AND ( "atypical social interactions" AND    ) ) OR ( Abstract: (   AND ( "atypical social interactions" AND    ) ) OR Full Text: (   AND ( "atypical social interactions" AND    ) ) ) ) OR ( ( Title: (   AND ( "social-communicative impairment" AND    ) ) OR ( Abstract: (   AND ( "social-communicative impairment" AND    ) ) OR Full Text: (   AND ( "social-communicative impairment" AND    ) ) ) ) OR ( ( Title: (   AND ( "social communicative impairment" AND    ) ) OR ( Abstract: (   AND ( "social communicative impairment" AND    ) ) OR Full Text: (   AND ( "social communicative impairment" AND    ) ) ) ) OR ( ( Title: (   AND ( "social-communicative deficit" AND    ) ) OR ( Abstract: (   AND ( "social-communicative deficit" AND    ) ) OR Full Text: (   AND ( "social-communicative deficit" AND    ) ) ) ) OR ( ( Title: (   AND ( "pragmatic language disability" AND    ) ) OR ( Abstract: (   AND ( "pragmatic language disability" AND    ) ) OR Full Text: (   AND ( "pragmatic language disability" AND    ) ) ) ) OR ( ( Title: (   AND ( "pragmatic disability" AND    ) ) OR ( Abstract: (   AND ( "pragmatic disability" AND    ) ) OR Full Text: (   AND ( "pragmatic disability" AND    ) ) ) ) OR ( ( Title: (   AND ( "pragmatic language dysfunction" AND    ) ) OR ( Abstract: (   AND ( "pragmatic language dysfunction" AND    ) ) OR Full Text: (   AND ( "pragmatic language dysfunction" AND    ) ) ) ) OR ( ( Title: (   AND ( "pragmatic dysfunction" AND    ) ) OR ( Abstract: (   AND ( "pragmatic dysfunction" AND    ) ) OR Full Text: (   AND ( "pragmatic dysfunction" AND    ) ) ) ) OR ( ( Title: (   AND ( "pragmatic language abnormalities" AND (   AND    ) ) ) OR ( Abstract: (   AND ( "pragmatic language abnormalities" AND (   AND    ) ) ) OR Full Text: (   AND ( "pragmatic language abnormalities" AND (   AND    ) ) ) ) ) OR ( ( Title: (   AND ( "pragmatic abnormalities" AND (   AND    ) ) ) OR ( Abstract: (   AND ( "pragmatic abnormalities" AND (   AND    ) ) ) OR Full Text: (   AND ( "pragmatic abnormalities" AND (   AND    ) ) ) ) ) OR ( ( Title: (   AND ( "pragmatic breakdown" AND    ) ) OR ( Abstract: (   AND ( "pragmatic breakdown" AND    ) ) OR Full Text: (   AND ( "pragmatic breakdown" AND    ) ) ) ) OR ( ( Title: (   AND ( "semantic pragmatic syndrome" AND    ) ) OR ( Abstract: (   AND ( "semantic pragmatic syndrome" AND    ) ) OR Full Text: (   AND ( "semantic pragmatic syndrome" AND    ) ) ) ) OR ( ( Title: (   AND ( "pragmatic and conversational deficit" AND    ) ) OR ( Abstract: (   AND ( "pragmatic and conversational deficit" AND    ) ) OR Full Text: (   AND ( "pragmatic and conversational deficit" AND    ) ) ) ) OR ( ( Title: (   AND ( "conversational disability" AND    ) ) OR ( Abstract: (   AND ( "conversational disability" AND    ) ) OR Full Text: (   AND ( "conversational disability" AND    ) ) ) ) OR ( ( Title: (   AND ( "verbal and non-verbal communication deficits" AND    ) ) OR ( Abstract: (   AND ( "verbal and non-verbal communication deficits" AND    ) ) OR Full Text: (   AND ( "verbal and non-verbal communication deficits" AND    ) ) ) ) OR ( ( Title: (   AND ( "pragmatic aphasia" AND    ) ) OR ( Abstract: (   AND ( "pragmatic aphasia" AND    ) ) OR Full Text: (   AND ( "pragmatic aphasia" AND    ) ) ) ) OR ( ( Title: (   AND ( "pragmatic dysphasia" AND    ) ) OR ( Abstract: (   AND ( "pragmatic dysphasia" AND    ) ) OR Full Text: (   AND ( "pragmatic dysphasia" AND    ) ) ) ) OR ( Title: (   AND ( "pragmatic and social difficulties" AND    ) ) OR ( Abstract: (   AND ( "pragmatic and social difficulties" AND    ) ) OR Full Text: (   AND ( "pragmatic and social difficulties" AND    ) ) ) ) ) ) ) ) ) ) ) ) ) ) ) ) ) ) ) ) ) ) ) ) ) ) ) ) ) ) ) |
| Filters: Publication Type = ( journal article , book chapter , unknown , book , dissertation , conference proceedings article    ) |
| Wednesday, 30 March 2022, 2308 Scholarly Works, 1977–2022 |

The present investigation centred on the employment of "pragmatic language impairment" and any synonymous terms to gauge the scope and evolution of research in this domain. Consequently, our search keywords did not encompass particular works confined by age, learner type, or language. When the aforementioned search strings and

our prior comprehension of the field were applied, an initial search on Google suggested the applicability of these search strings to probing knowledge concerning PLI. Per the Lens database, Figure 1 displays concepts that are synonymous with PLI.

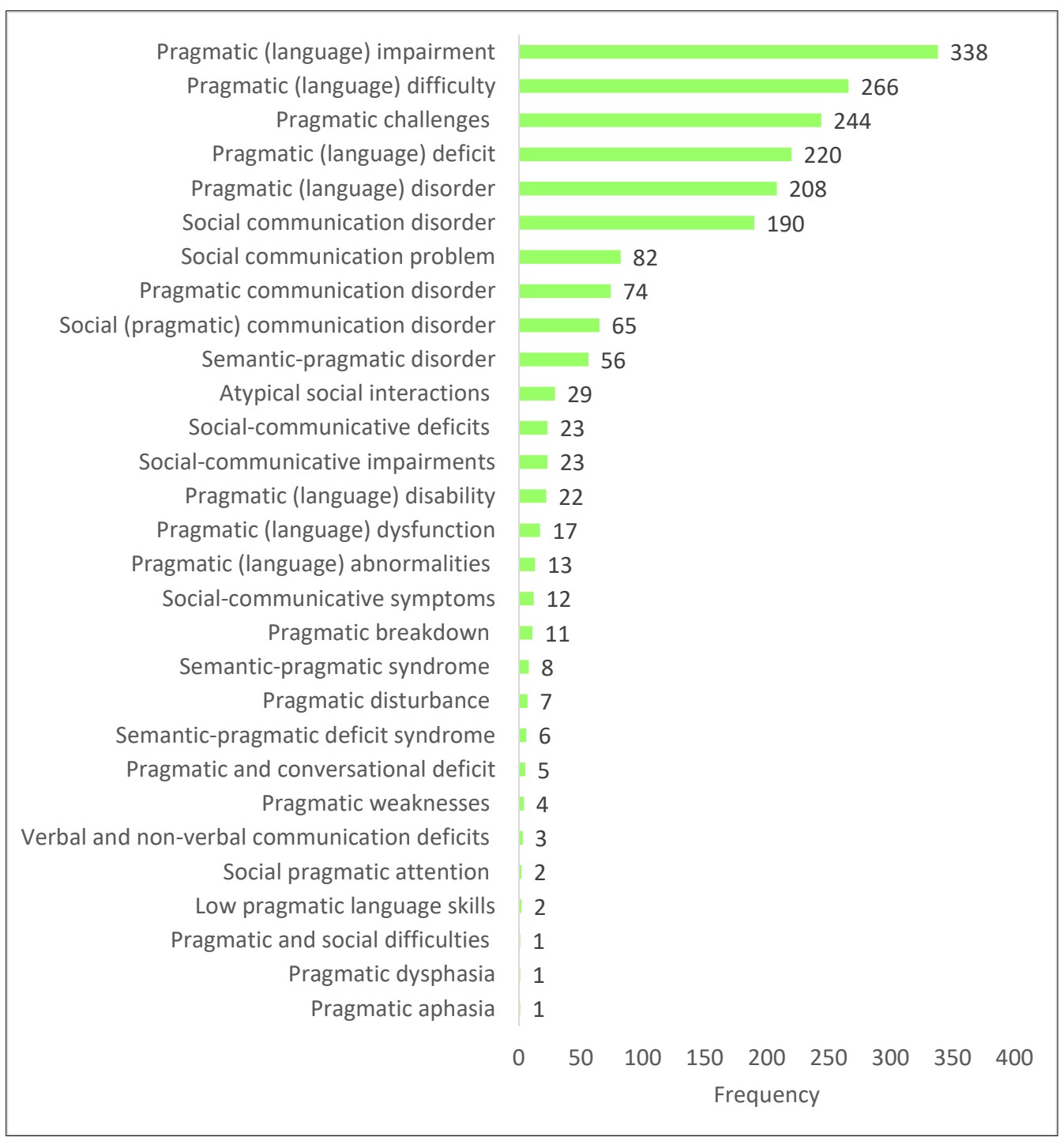

**Figure 1.** Synonymous Concepts for Pragmatic Language Impairment in Lens Database.

*2.4. Data Analysis*

Several measures were undertaken prior to initiating data analysis. Initially, Scopus data were exported in three formats: Excel sheets for bibliometric analysis, RIS for CiteSpace, and CSV for VOSviewer. In accordance with CiteSpace's prerequisites, the RIS file was converted to WOS format. Moreover, WOS data were procured in two formats:

text documents converted to Excel spreadsheets for bibliometric analysis, and plain text documents for CiteSpace and VOSviewer. Finally, Lens data were acquired in two formats: CSV for bibliometric analysis, and full record CSV for VOSviewer.

Duplicate documents were eliminated in CiteSpace and Mendeley before commencing the CiteSpace analysis. Excel was employed to carry out the bibliometric analysis. Citation reports were generated using Excel and subsequently converted to figures.

Scientometric analyses were executed with default configurations in both software tools. For each of the three databases, a network visualisation, an overlay visualisation, and a density visualisation were generated. The Scopus and WOS results were analysed three times: by author keywords, by source, and by cited author. For Lens, four analyses were conducted: co-occurrence analysis by author keywords, citation analysis by author, citation analysis by source, and citation analysis by document. Among the CiteSpace data for Scopus and WOS, the following analyses were performed: co-citations by document (references), co-citations by cited authors, and occurrence (keywords). Diverse summaries were produced, encompassing narrative summaries, cluster summaries, visual maps, and burst tables.

## 3. Results

### 3.1. Result Overview

The outcomes of this study are organised into two sections. First, we present bibliometric indicators for PLI, which are derived from data obtained from Scopus, WOS, and Lens. These indicators encompass publications by year, top 10 countries, universities, journals, publishers, subject/research areas, and authors. Second, we introduce scientometric indicators for the evolution of PLI, analysed using VOSviewer and CiteSpace software. These indicators include citation, co-citation, and co-occurrence measures.

### 3.2. Bibliometric Indicators for the Study of Pragmatic Language Impairment

3.2.1. Overview of PLI Studies from Scopus, Web of Science, and Lens

A combined total of 1069 PLI documents were retrieved from Scopus, 475 from WOS, and 2308 from Lens. The data period for each database spanned the periods between 1977 and 2022, 1985 and 2022, and 1977 and 2022, respectively. Scopus contained 812 articles, 107 review articles, 88 book chapters, 10 books, and 52 conference papers. WOS documents included 428 articles, 29 reviews, 23 book chapters, 6 early access papers, and 31 proceedings papers. Lens documents comprised 1723 articles, 204 unidentified articles, 185 book chapters, 68 books, 40 dissertations, and 33 conference proceedings. The majority of these documents were in English, with a few in Spanish, French, German, Italian, Korean, and Polish. Given that the analysis is based on title, keywords, abstract, and references, all of these encompass English language content. The authors decided to include these data to prevent bias towards English-language publications.

Figure 2A–C showcase the production length for the three databases by year. PLI knowledge production has experienced significant growth in recent years, reaching its zenith in Scopus in 2017 with 89 publications, in WOS with 39 publications in 2020, and in Lens in 2016 with 201 publications. Scopus publications range from 1 to 89, WOS publications from 1 to 39, and Lens publications from 1 to 201. The previous year's publications were the lowest for all databases. As a result, knowledge production related to PLI has increased over the past two decades.



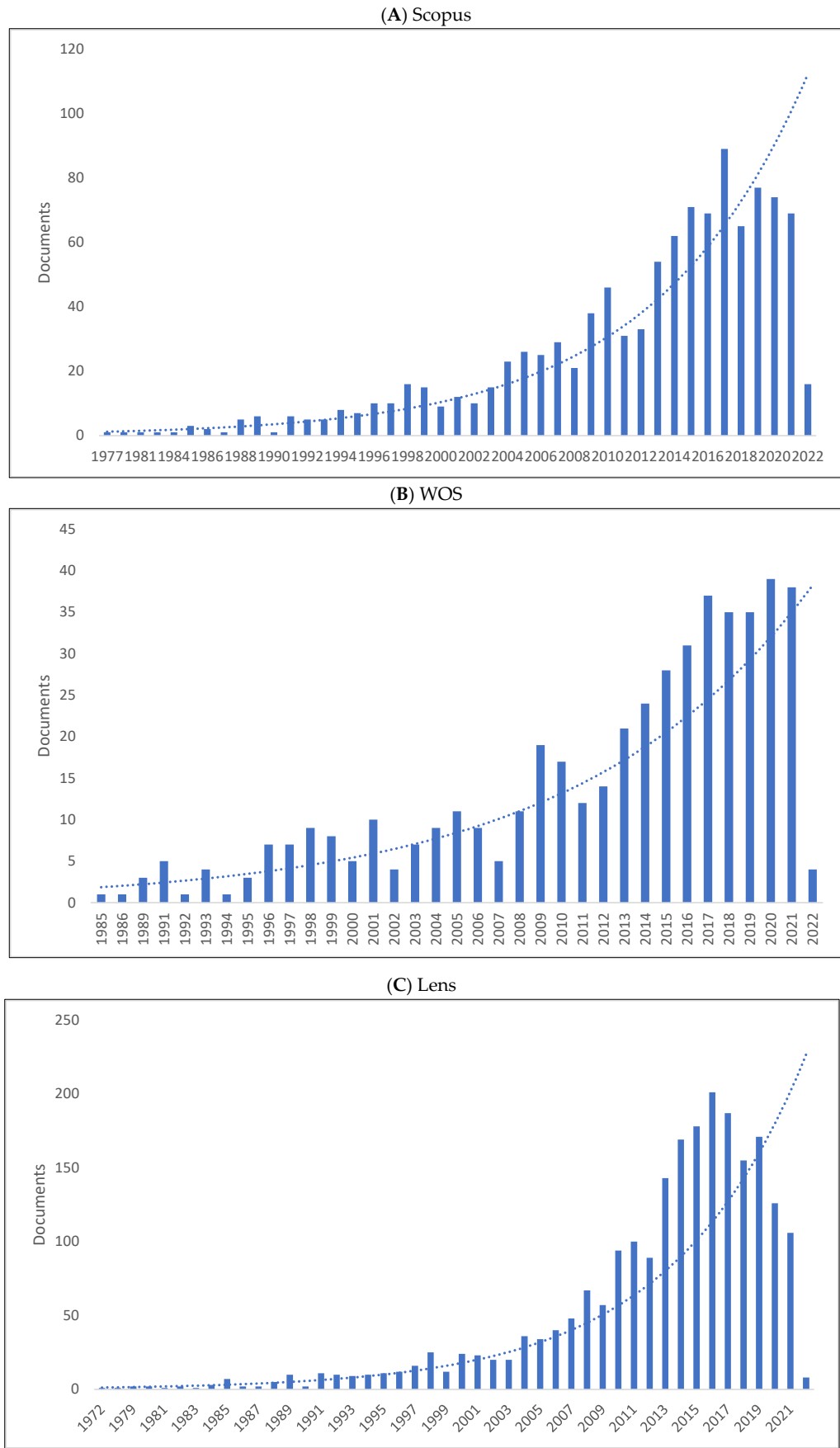

**Figure 2.** Knowledge Production Size in Pragmatic Language Impairment by Year.

### 3.2.2. Production of Pragmatic Language Impairment Research by Country and University

Figure 3A–C display the top ten producing countries for PLI knowledge. Both Scopus and WOS rank the US first, while Lens positions the UK first. Other countries are located in Europe, Australia, and North America, with China as the sole country from the rest of the world.

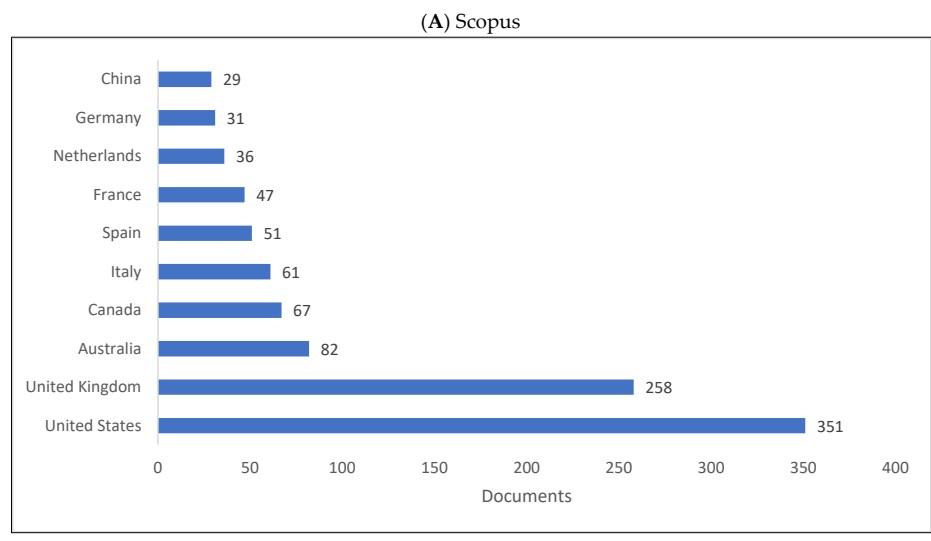

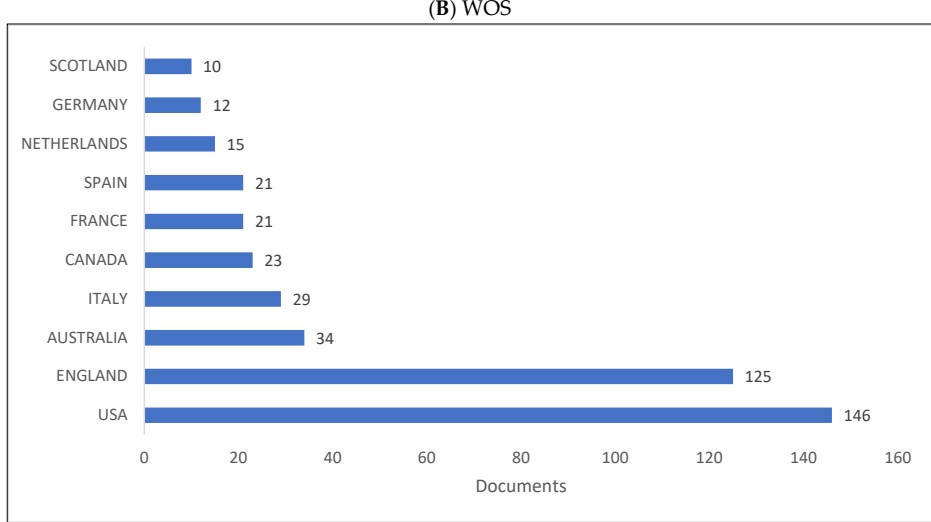

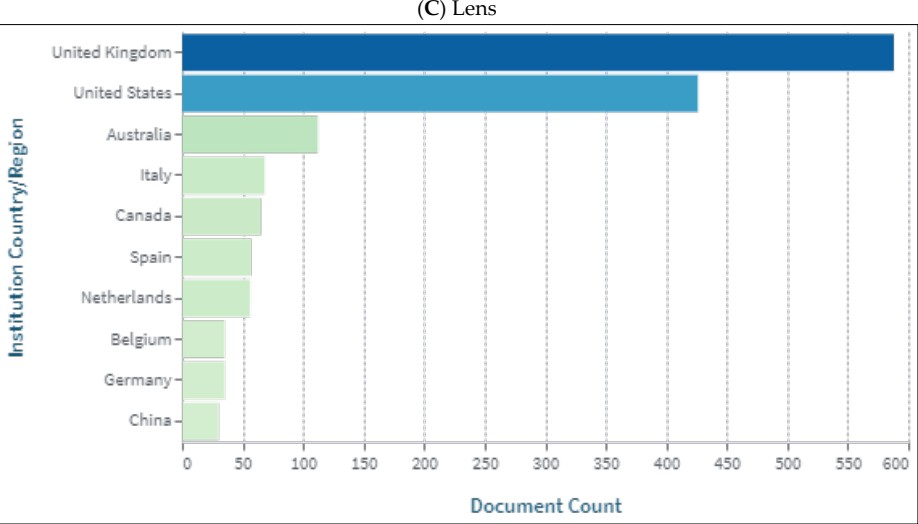

**Figure 3.** Knowledge Production Regarding Pragmatic Language Impairment by Country.

Figure 4A–C illustrate the top 10 universities and/or research centres generating knowledge in PLI. The three databases indicate that UK universities dominate the list. PLI research is primarily published by researchers at the University of Manchester and University College London.

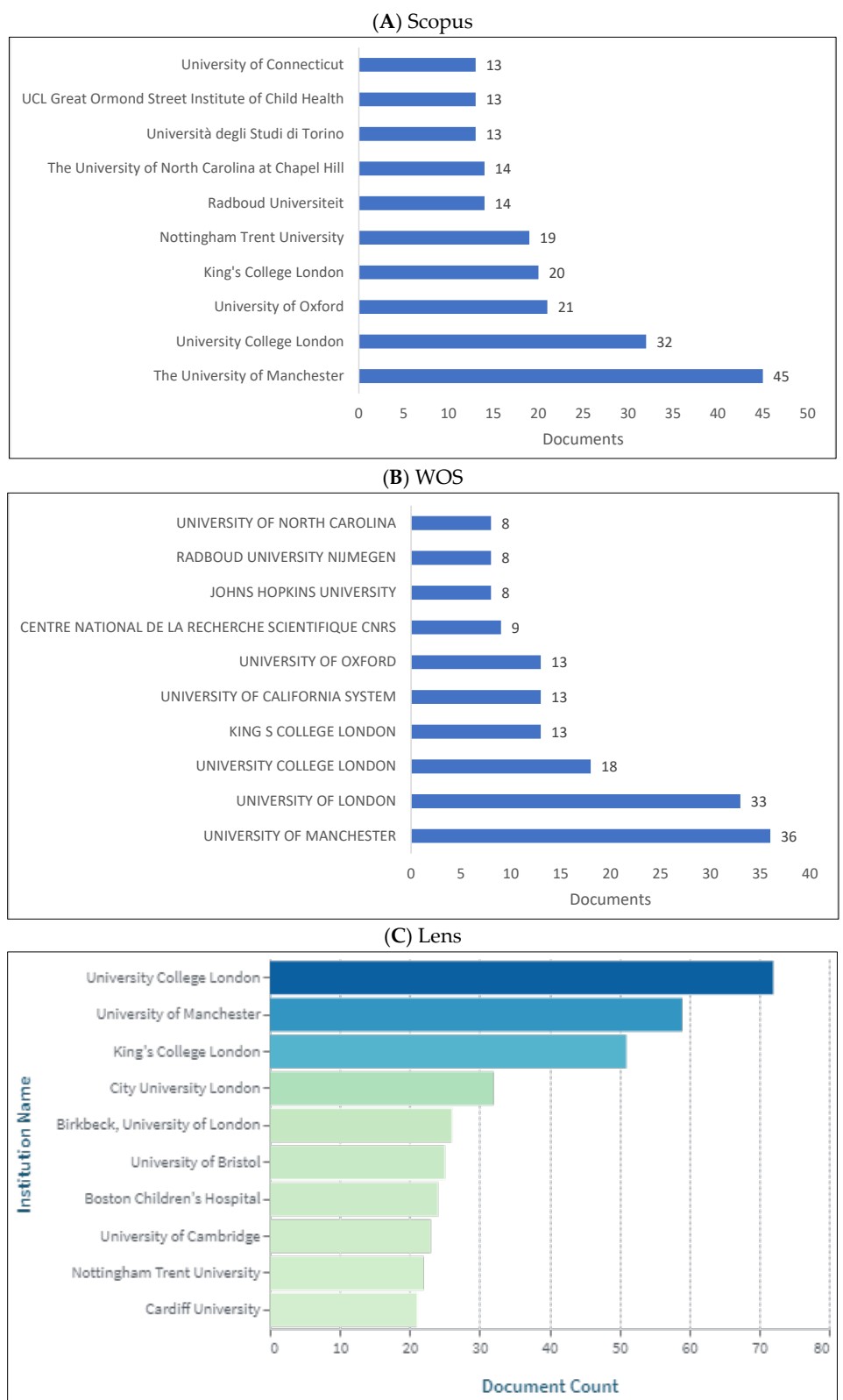

**Figure 4.** Knowledge Production of Pragmatic Language Impairment by University.

### 3.2.3. Production of Pragmatic Language Impairment Research by Journal and Publisher

Figure 5A–D list the top ten journals publishing research in PLI. PLI research is closely related to health sciences, psychology, and neuroscience, so the journals concentrate on these research areas. The majority of these journals address speech and language disorders, autism, psychology, neuroscience, clinical linguistics, and neurolinguistics. An extended list of journals based on publishers is shown in Figure 5D, which includes one journal called the *Journal of Pragmatics*; the rest are related to psychology or neuroscience.

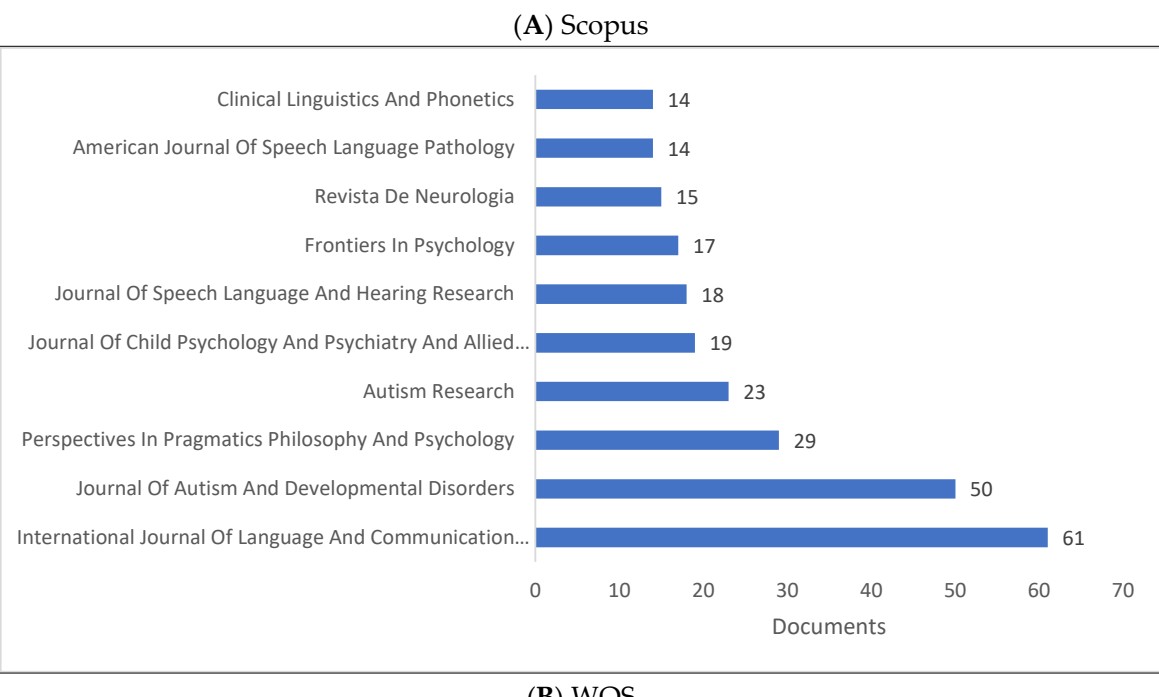

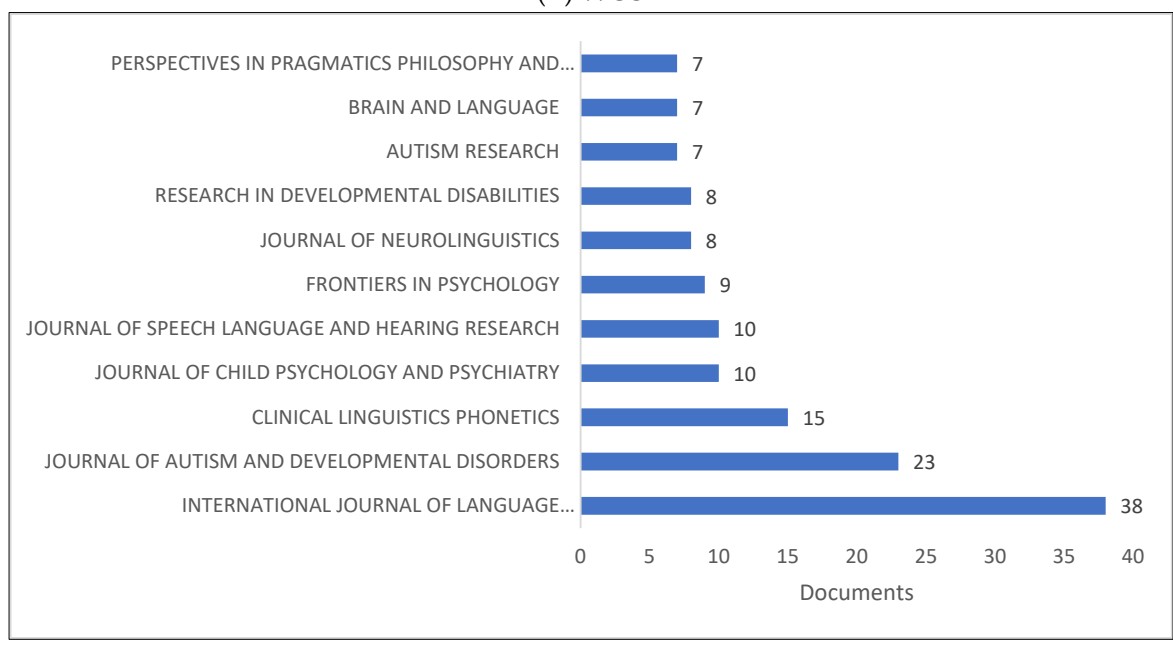

**Figure 5.** *Cont.*

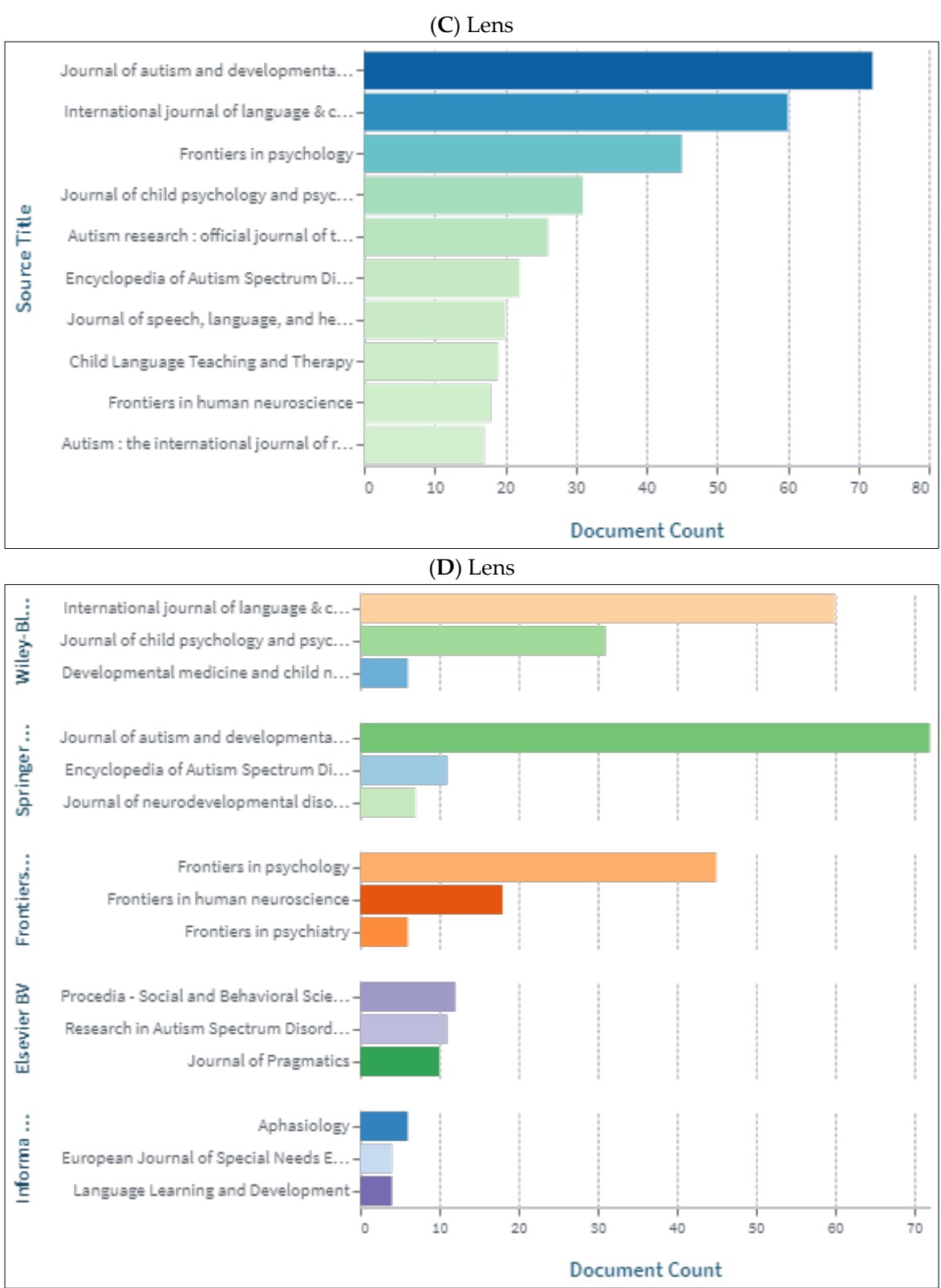

**Figure 5.** Knowledge Production of Pragmatic Language Impairment by Journal.

Figure 6A,B exhibit a list of the top 10 knowledge publishers in PLI. As Scopus does not include publisher information, these lists are limited to WOS and Lens. Both databases rank Elsevier and Wiley as the most prolific publishers, but the remaining publishers vary between them. For instance, Springer Nature ranks 4th in WOS but 3rd in Lens.

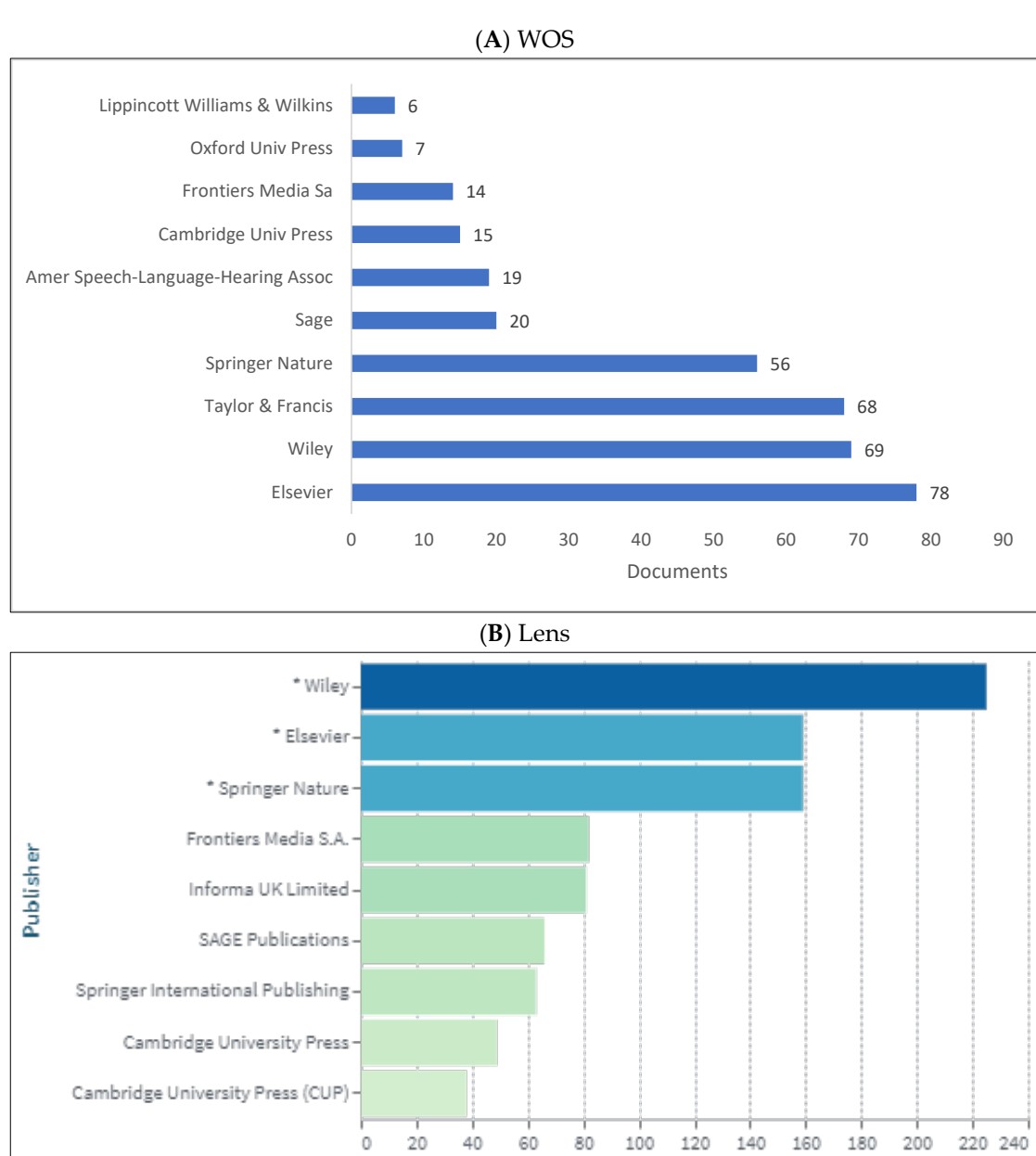

**Figure 6.** Knowledge Production of Pragmatic Language Impairment by Publisher. The asterisk (*) symbol indicates top three publishers for research related to PLI.

### 3.2.4. Production of Pragmatic Language Impairment by Research Area, Keywords, and Co-Occurrence

PLI studies are considered a subfield of pragmatics, but they are also integrated with numerous other fields, as demonstrated in Figure 7A–C. According to Figure 7A, medicine, social sciences, psychology, and arts and humanities are the top four subject areas publishing in PLI. Figure 7B reveals that the top four research areas in PLI are psychology, linguistics, rehabilitation, and audiology–speech–language pathology. Figure 7C confirms this, introducing psychology, developmental psychology, autism, and cognitive psychology as the top four fields of study in PLI. More specific fields related to PLI are displayed in Lens (e.g., pragmatics, language use, theory of mind, and non-verbal communication).

**(A)** Scopus

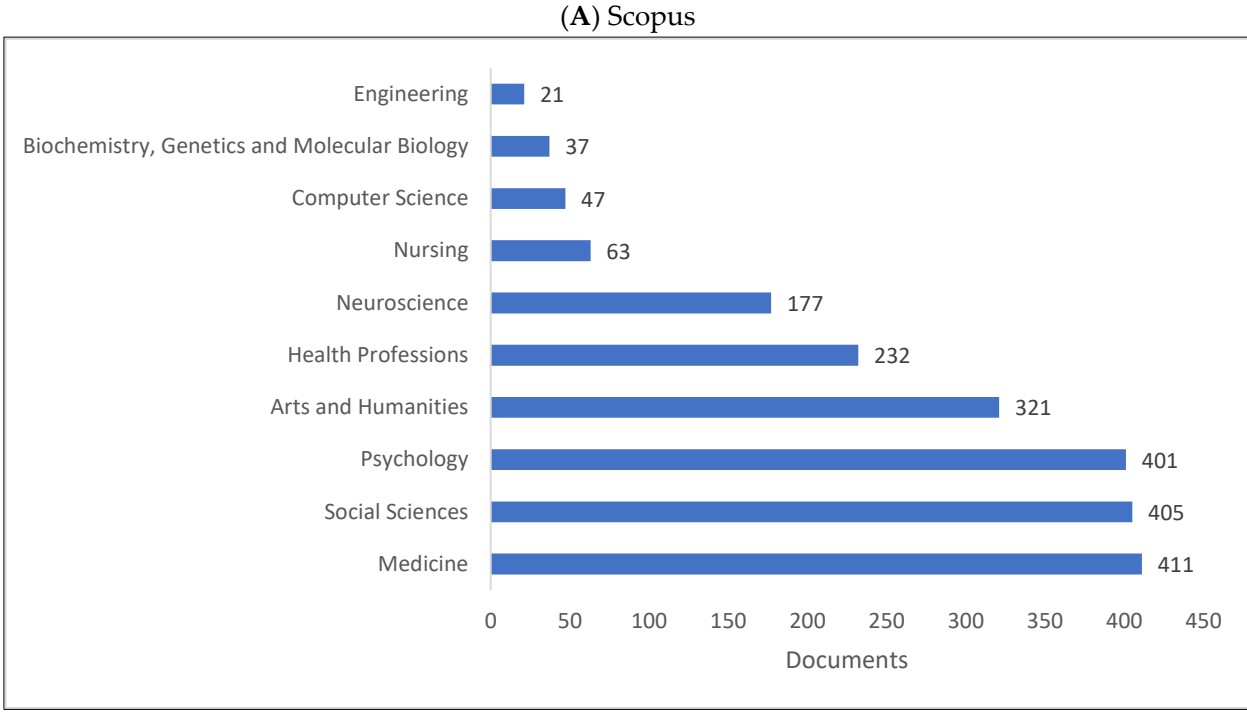

**(B)** WOS

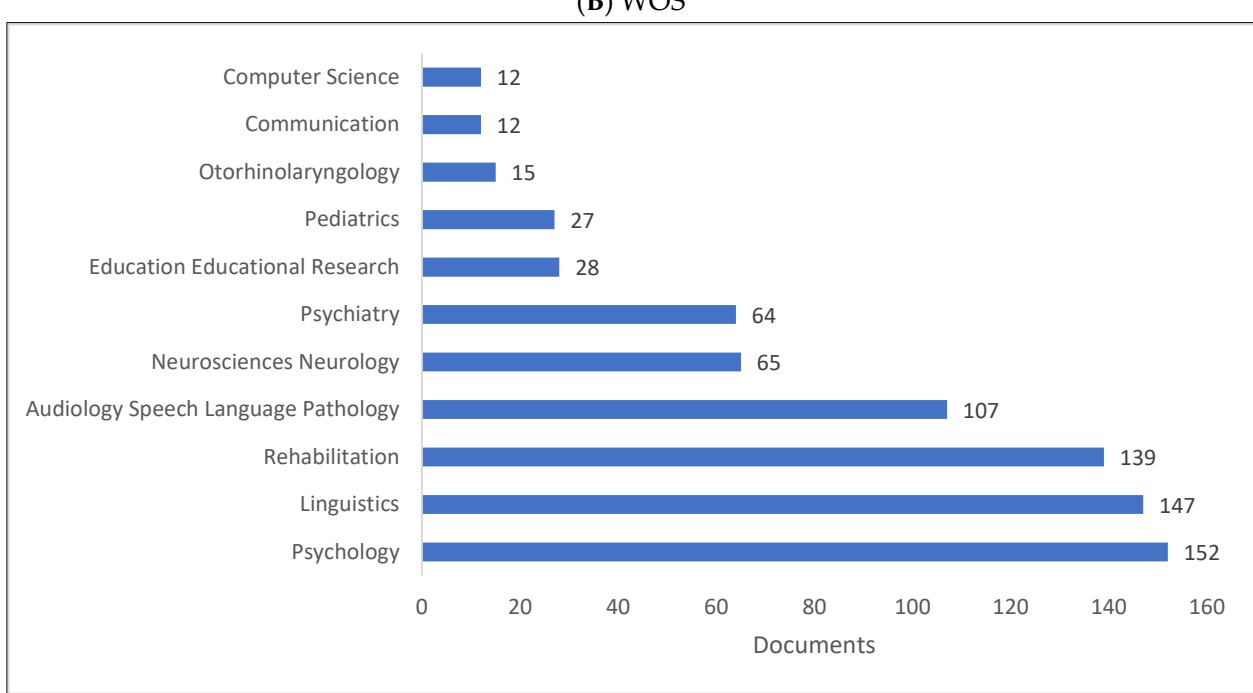

**Figure 7.** *Cont.*

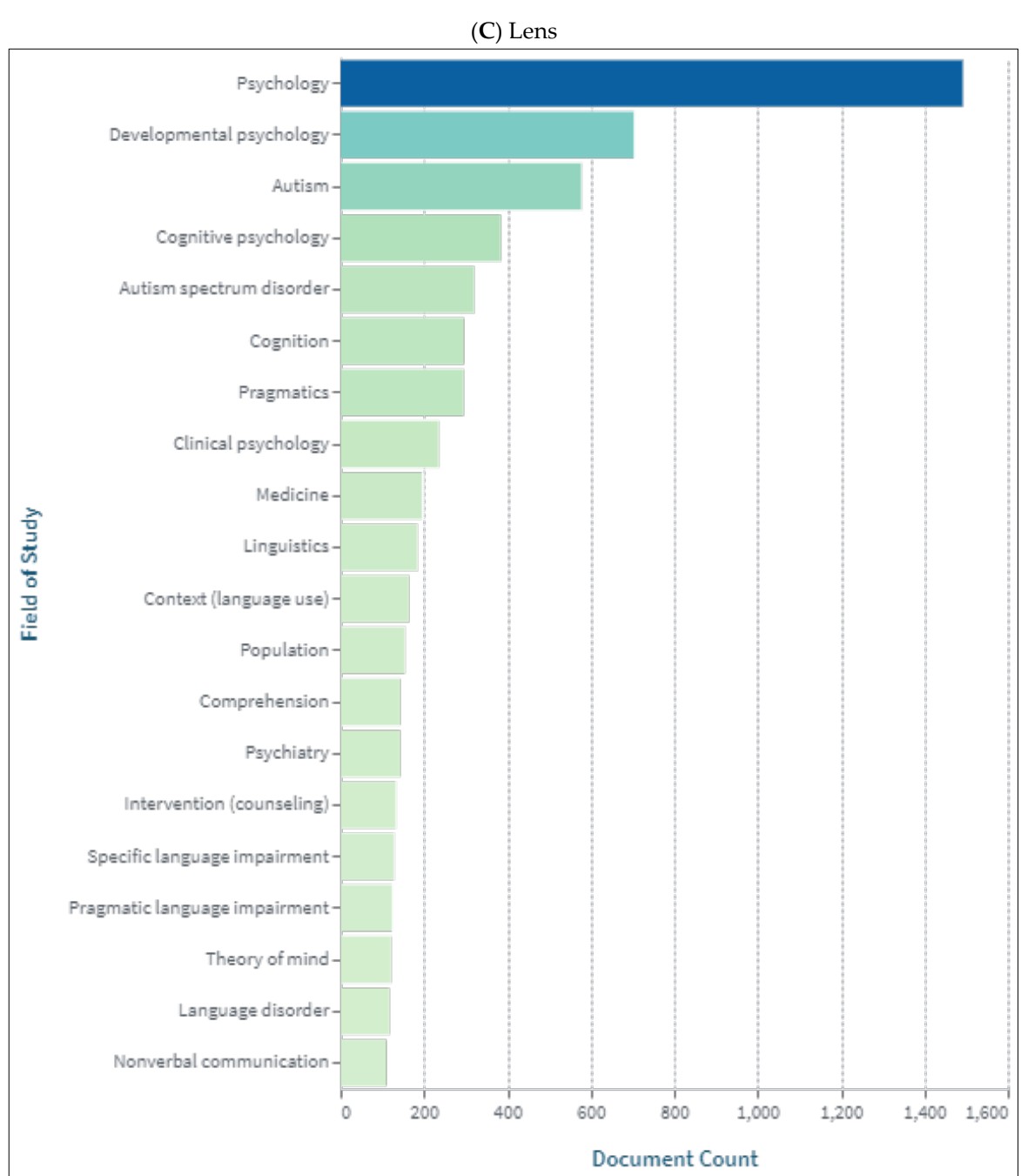

**Figure 7.** Knowledge Production of Pragmatic Language Impairment by Research Area.

3.2.5. Production of Pragmatic Language Impairment by Authors

Undoubtedly, the contributions to the field of PLI extend beyond a specific set of authors, as even a single article can contribute to the domain. In contrast, our aim was to analyse the authors who have generated a larger body of knowledge related to PLI, as demonstrated in Figure 8A–C. In all databases, Adams [35] is the first author followed by either Bishop [10] or Cummings [59].

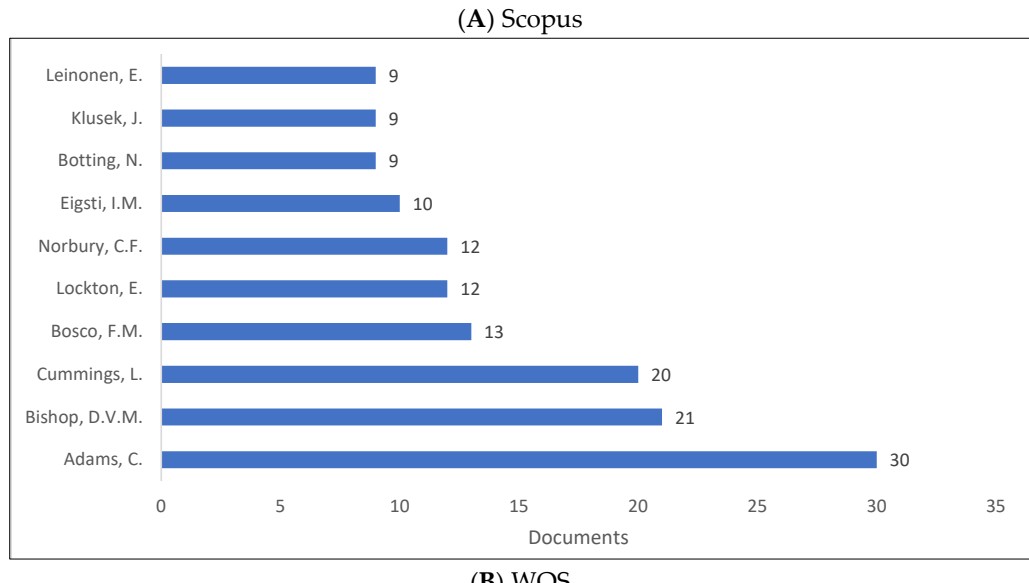

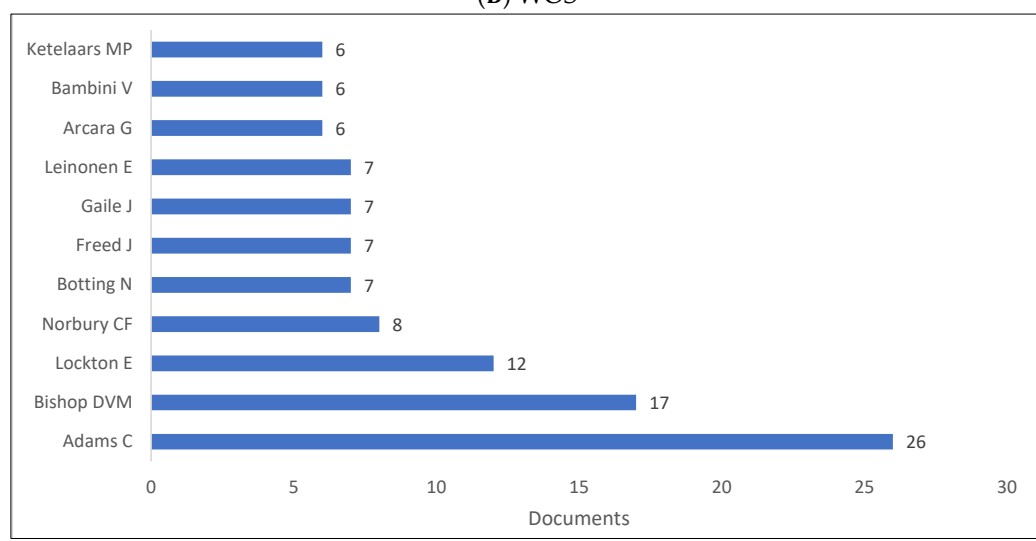

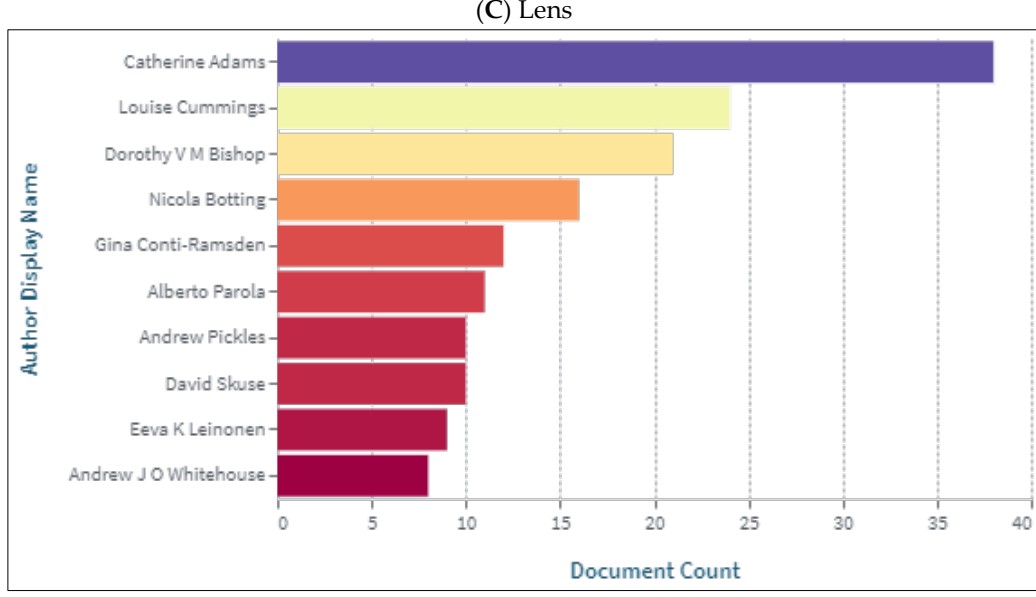

**Figure 8.** Knowledge Production of Pragmatic Language Impairment by Author.

*3.3. Scientometric Indicators for the Study of Pragmatic Language Impairment*
Overview of PLI Studies from Scopus, Web of Science, and Lens

In this section, we present the outcomes of the scientometric analysis conducted on the data retrieved from the Scopus, WOS, and Lens databases, with a particular focus on emphasizing the influence of specific concepts, authors, references, and emerging trends within the field of PLI.

CiteSpace was utilised to visualise the top keywords with the most robust citation bursts from Scopus and WOS (Figure 9A,B). The green line represents the period encompassing all research, while a red line signifies the start and end of the burst period. Scopus identifies the strongest citation burst for "language disability" (26.51) between 1979 and 2012 and "conversational characteristics" (9.56) between 1991 and 2006 for WOS. The citation burst varies depending on the database. In WOS, we observed pragmatic disorder, PLI, and social communication disorder, whereas in Scopus, only social communication disorder is noted. In WOS, social communication disorder first appeared in 2012, while in Scopus, it emerged in 2015.

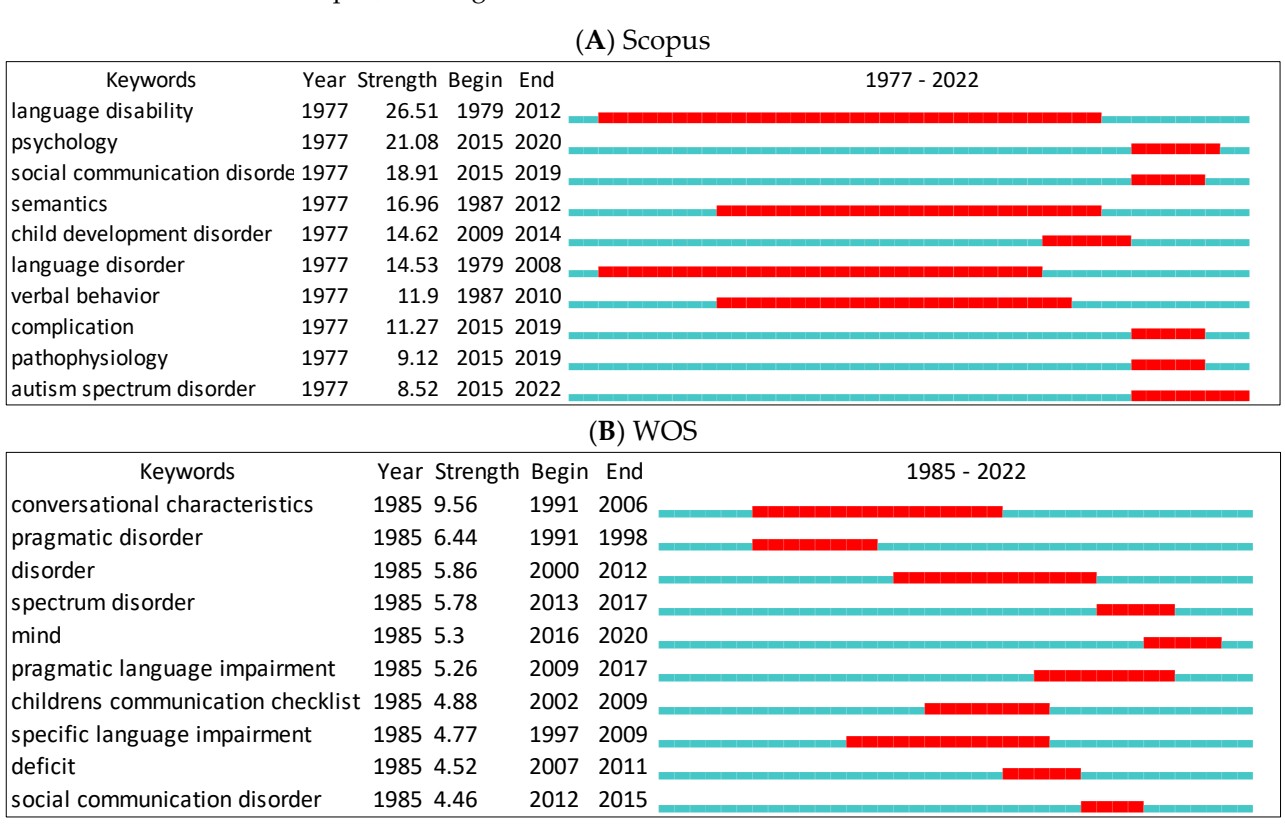

**Figure 9.** Top 10 Keywords with the Strongest Citation Bursts.

Keyword co-occurrence is another crucial aspect. We created three visual network maps to depict the most frequently used keywords in PLI across the three databases using VOSviewer (Figure 10A–C). The colours represent distinct research directions within the study of PLI. Pragmatics-related topics are displayed in green, communication-related topics in blue, and autism-related topics in red (see Figure 10A). The colours vary depending on the database. In Figure 11B, green signifies topics connected to social communication disorders, purple denotes topics related to the theory of mind, and red indicates topics associated with assessment and diagnosis. In Figure 10C, the sky-blue region encompasses keywords relevant to aphasia and PLI, while the orange region represents social interaction.

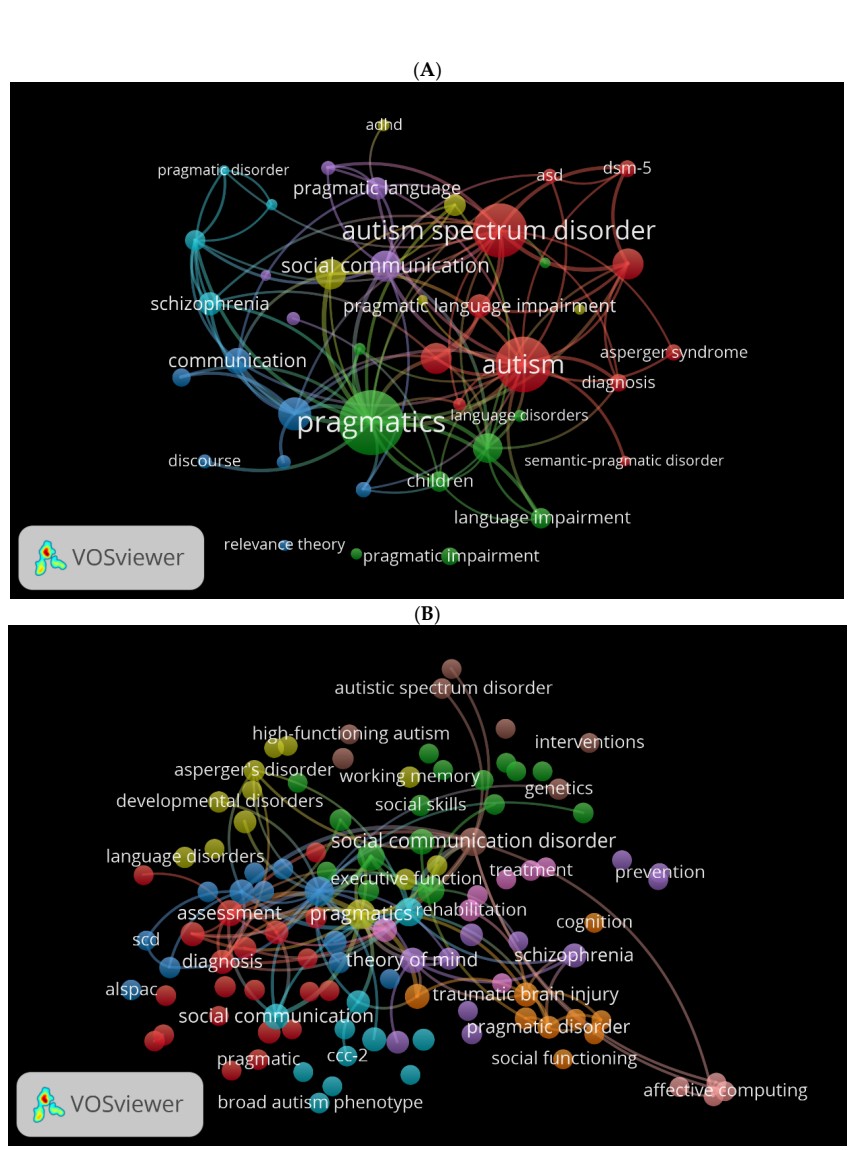

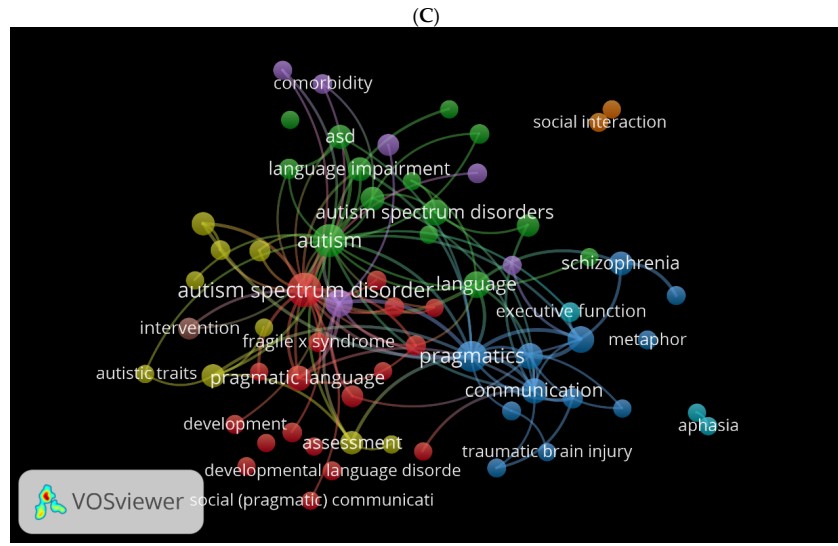

**Figure 10.** Co-occurrence by Author Keywords Network Visualisation. (**A**) (Scopus) Pragmatics-related topics (Green), Communication-related topics (Blue), and Autism-related topics (Red). (**B**) (WOS) Social Communication Disorders (Green), Theory of Mind (Purple), and Assessment and Diagnosis (Red). (**C**) (Lens) Aphasia and PLI-related keywords (Sky-Blue), Social Interaction-related keywords (Orange).

(**A**) Scopus

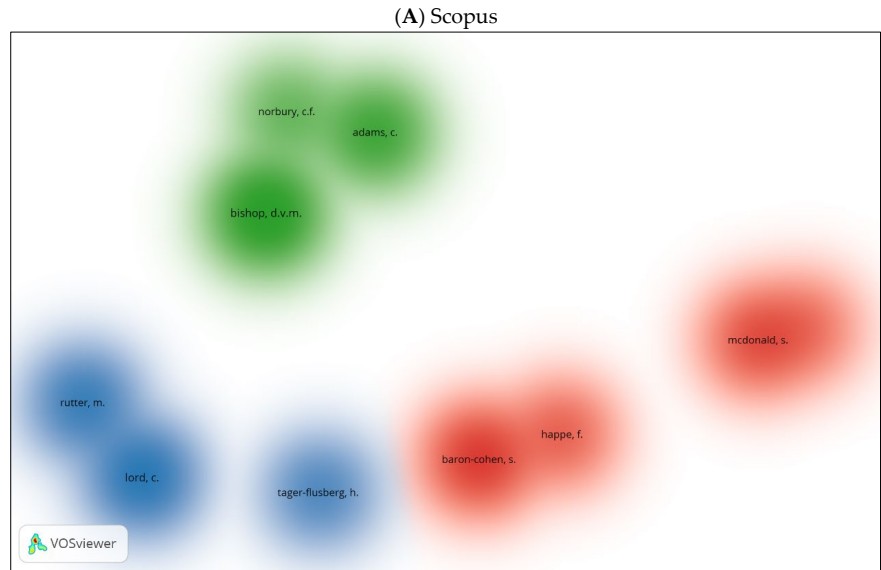

(**B**) WOS

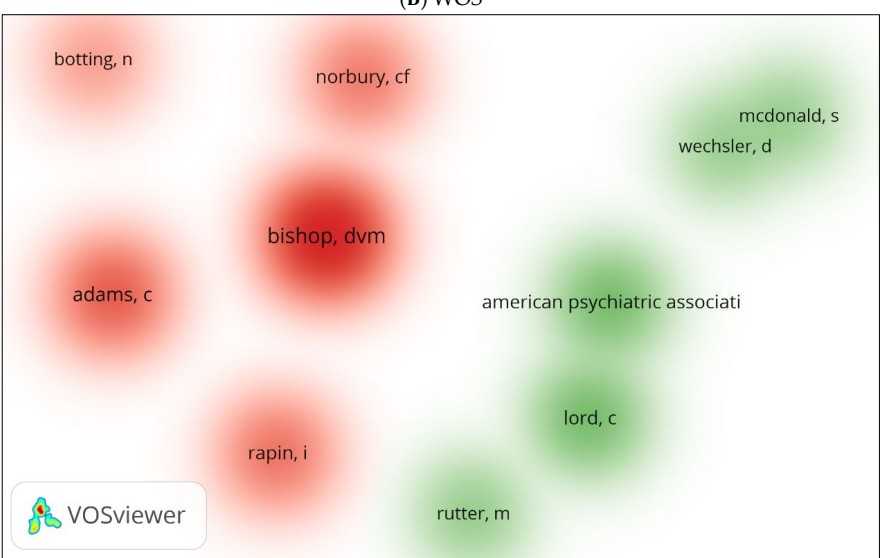

(**C**) Lens: Citation by Author Density Visualisation

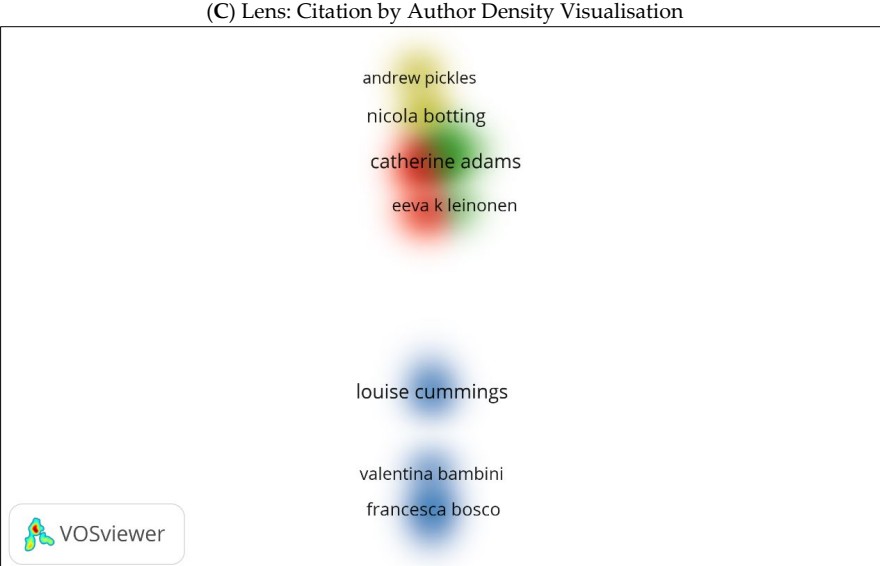

**Figure 11.** Co-citation by Cited Author Density Visualisation.

We generated three visual network maps for co-citation and citation by author using VOSviewer (Figure 11A–C). Each colour represents a co-citation or citation network. Co-citations and citations increase in size as the circle gets larger. Co-citations and citations by the same authors can be found in all three databases. The list includes Adams [35], Bishop [10], Cummings [59], and Bambini [60].

We generated three visual network maps utilising VOSviewer to represent co-citation and citation by source (Figure 12A–C). Each colour corresponds to a network of co-citations or citations for sources. A larger circle size signifies a higher co-citation or citation frequency. As per Figure 12A, sources such as the *Journal of Autism*, *Brain and Language*, and *Neuropsychologia* appear to be the most frequently cited. Figure 12B presents similar outcomes using the WOS database, featuring more prominent journals (e.g., *Journal of Speech and Hearing Research*). The citation network for journals is depicted in Figure 12C, which includes the *International Journal of Language and Communication Disorders, Frontiers in Psychiatry*, and others.

(**A**) Scopus

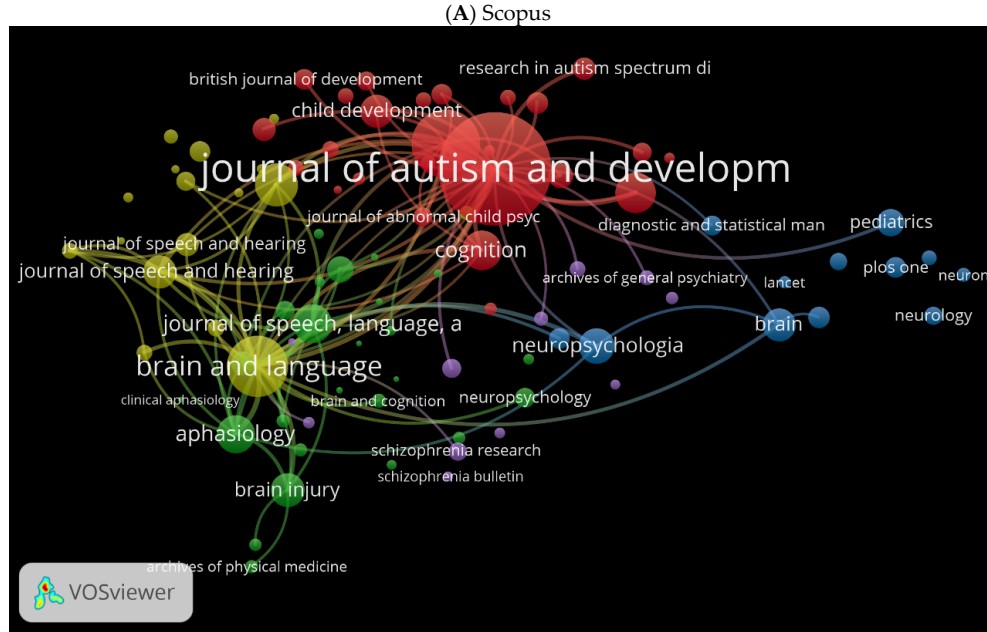

(**B**) WOS

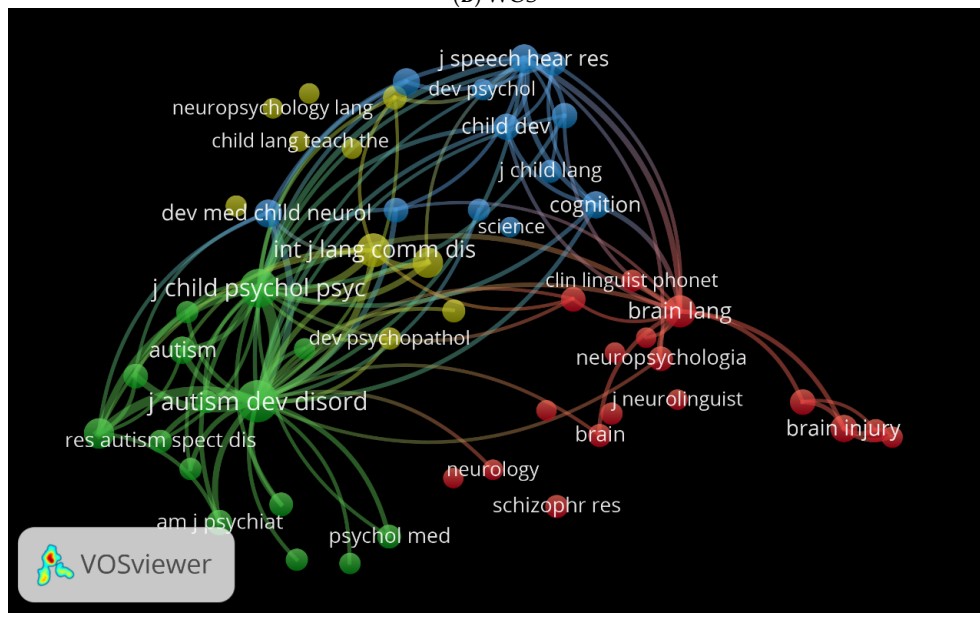

**Figure 12.** *Cont.*

(**C**) Lens: Citation by Source

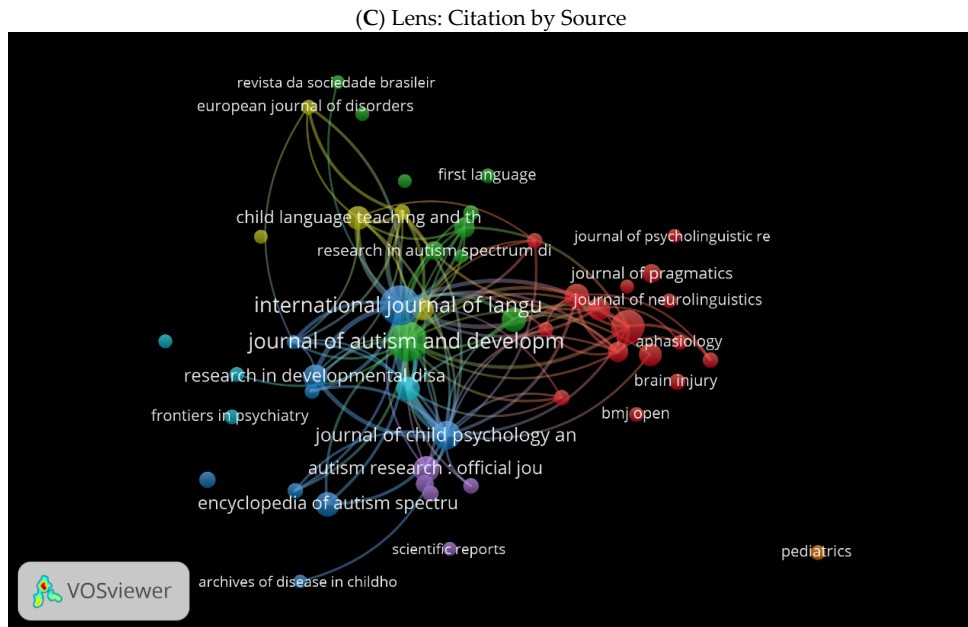

**Figure 12.** Co-citation by Source Network Visualisation.

Utilizing bibliometric data from Scopus, WOS, and Lens, we compiled the top 10 cited works in each database. After eliminating duplicates, these three lists were merged (Table 3). The ranking and list of cited works vary depending on the database used. For example, "Development of the Children's Communication Checklist (CCC): A method for assessing qualitative aspects of communicative impairment in children" holds the top position in Scopus with 442 citations, ranks second in WOS with 393 citations, and is third in Lens, but with 592 citations.

**Table 3.** Top Cited Documents of PLI Based on Citation Reports from Scopus, WOS and Lens.

| No. | Source Title | Citation | Citations |
|---|---|---|---|
| 1 | A Behavioral Comparison of Male and Female Adults with High Functioning Autism Spectrum Conditions | [61] | 410 |
| 2 | Acquired "theory of mind" impairments following stroke | [62] | 339 |
| 3 | Action Anticipation Through Attribution of False Belief by 2-Year-Olds | [63] | 679 |
| 4 | Development of the Children's Communication Checklist (CCC): A method for assessing qualitative aspects of communicative impairment in children | [9] | 592 |
| 5 | Exploring the borderlands of autistic disorder and specific language impairment: A study using standardised diagnostic instruments | [14] | 322 |
| 6 | Gaze cueing of attention: visual attention, social cognition, and individual differences. | [64] | 992 |
| 7 | Health effects of housing improvement: Systematic review of intervention studies | [65] | 251 |
| 8 | Inferential processing and story recall in children with communication problems: a comparison of specific language impairment, pragmatic language impairment and high-functioning autism | [13] | 181 |
| 9 | Narrative skills of children with communication impairments | [22] | 412 |
| 10 | Neural basis of irony comprehension in children with autism: The role of prosody and context | [66] | 248 |
| 11 | Phase 2 of CATALISE: a multinational and multidisciplinary Delphi consensus study of problems with language development: Terminology | [67] | 532 |
| 12 | Precise minds in uncertain worlds: Predictive coding in autism | [68] | 409 |
| 13 | Preliminary communication social language use in parents of autistic individuals | [69] | 239 |
| 14 | Psychosis and autism as diametrical disorders of the social brain | [70] | 446 |
| 15 | Sensitivity and specificity of proposed DSM-5 diagnostic criteria for autism spectrum disorder | [71] | 261 |
| 16 | Social and pragmatic deficits in autism: Cognitive or affective? | [72] | 578 |
| 17 | Social Difficulties and Victimisation in Children with SLI at 11 Years of Age | [73] | 221 |
| 18 | The extent to which psychometric tests differentiate subgroups of children with SLI | [74] | 209 |
| 19 | The eye contact effect: mechanisms and development | [75] | 552 |
| 20 | The screening and diagnosis of autistic spectrum disorders | [76] | 616 |

**Table 3.** *Cont.*

| No. | Source Title | Citation | Citations |
|---|---|---|---|
| 21 | Using a parental checklist to identify diagnostic groups in children with communication impairment: a validation of the Children's Communication Checklist–2 | [77] | 175 |
| 22 | Weak coherence, no theory of mind, or executive dysfunction? Solving the puzzle of pragmatic language disorders | [36] | 235 |

*3.4. Impact of Research on Pragmatic Language Impairment by Clusters, Citation Counts, Citation Bursts, Centrality, and Sigma*

3.4.1. Clusters

The network is divided into 16 co-citation clusters in the Scopus data (See Table 4). The largest cluster (#0) has 171 members and a silhouette value of 0.615. It is labelled as children's communication checklist by LLR, social communication disorder by LSI, and behavioural problems (2.48) by MI. The most relevant citer to the cluster is Perkins [78] "Pragmatic Impairment".

**Table 4.** Summary of the Largest Clusters for Pragmatic Language Impairment.

| Cluster ID | Size | Silhouette | Label (LSI) | Label (LLR) | Label (MI) | Average Year |
|---|---|---|---|---|---|---|
| | | | | Scopus | | |
| 0 | 171 | 0.615 | social communication disorder | children's communication checklist (766.8, $1.0 \times 10^{-4}$) | behavioural problem (2.48) | 2007 |
| 1 | 157 | 0.811 | traumatic brain injury | traumatic brain injury | behavioural problem (1.45) | 2010 |
| 2 | 107 | 0.635 | autism spectrum disorder | autism spectrum disorder | behavioural problem (1.94) | 2011 |
| 3 | 87 | 0.866 | autism spectrum disorder | contingent relationship | behavioural problem (0.19) | 1988 |
| 4 | 71 | 0.779 | autism spectrum disorder | Asperger's syndrome | belief report (1.07) | 2007 |
| 5 | 57 | 0.92 | autism spectrum disorder | DSM-5 diagnostic criteria | behavioural problem (0.38) | 2010 |
| | | | | WOS | | |
| 0 | 164 | 0.638 | autism spectrum disorder | traumatic brain injury | diagnostic observation schedule score (3.04) | 2006 |
| 1 | 96 | 0.835 | inferential meaning | inferential meaning | assessing qualitative aspect (0.9) | 1996 |
| 2 | 94 | 0.897 | traumatic brain injury | traumatic brain injury | irony elaboration (1.1) | 2014 |
| 3 | 90 | 0.871 | Asperger's syndrome | Asperger's syndrome | diagnostic observation schedule score (0.52) | 2002 |

The network is divided into 12 co-citation clusters in the WOS data (See Table 4). The largest 4 clusters are summarised as follows. The largest cluster (#0) has 164 members and a silhouette value of 0.638. It is labelled as traumatic brain injury by LLR, autism spectrum disorder by LSI, and diagnostic observation schedule score (3.04) by MI. The most relevant citer to the cluster is Amoretti [39] "The DSM-5 introduction of the social (pragmatic) communication disorder as a new mental disorder: a philosophical review".

3.4.2. Citation Counts

In Scopus, the top-ranked item by citation counts is Bishop [79] in Cluster #0, with citation counts of 243. The second one is Lord [80] in Cluster #2, with citation counts of 214. In WOS, the top-ranked item by citation counts is Bishop [81] in Cluster #0, with citation

counts of 184. The second one is the American Psychiatric Association [6] in Cluster #0, with citation counts of 128. The details can be found in Table 5.

**Table 5.** Citation Counts for Pragmatic Language Impairment Using Scientometric Analysis.

| WoS | | | Scopus | | |
|---|---|---|---|---|---|
| Citation | Reference | Cluster ID | Citation | Reference | Cluster ID |
| 184 | Bishop [81] | 0 | 243 | Bishop [20] | 0 |
| 128 | American Psychiatric Association [6] | 0 | 214 | Lord [80] | 2 |
| 118 | Adams [81] | 0 | 170 | Baron-Cohen [72] | 4 |
| 92 | Rapin [7] | 0 | 155 | Adams [81] | 0 |
| 91 | Lord [82] | 0 | 145 | [Anonymous], 1979 | 8 |
| 83 | Norbury [83] | 0 | 139 | Rapin [17] | 0 |
| 54 | Rutter [84] | 0 | 126 | Norbury [77] | 0 |
| 53 | Wechsler [85] | 1 | 122 | Tager-Flusberg [86] | 0 |
| 47 | Botting [87] | 0 | 109 | Rutter [88] | 0 |
| 47 | Tager-Flusberg [86] | 0 | 103 | Wechsler [85] | 0 |

### 3.4.3. Bursts

In Scopus, the top-ranked item by bursts is Rapin [17] in Cluster #0, with bursts of 14.95. The second one is Cummings [89] in Cluster #1, with bursts of 11.68. In the WOS, the top-ranked item by bursts is [Anonymous] (2014) in Cluster #2, with bursts of 11.04. The second one is Swineford [90] in Cluster #4, with bursts of 7.88. See Table 6 and Figure 13A–D for more detail.

**Table 6.** Bursts Ranking for Pragmatic Language Impairment Using Scientometric Analysis.

| WoS | | | Scopus | | |
|---|---|---|---|---|---|
| Burst | Reference | Cluster ID | Burst | Reference | Cluster ID |
| 11.04 | [Anonymous], 2014 | 2 | 14.95 | Rapin [17] | 0 |
| 7.88 | Swineford [90] | 4 | 11.68 | Cummings [89] | 1 |
| 7.62 | Rapin [17] | 0 | 11.16 | Mctear [91] | 0 |
| 7.44 | Mandy [92] | 4 | 10.86 | Botting [87] | 0 |
| 7.27 | Dunn [93] | 1 | 9.93 | Douglas [94] | 1 |
| 6.91 | Cummings [95] | 2 | 8.91 | Conti-Ramsden [96] | 0 |
| 6.59 | Mcdonald [97] | 2 | 8.79 | Bosco [98] | 1 |
| 6.54 | Semel [99] | 0 | 8.56 | Shields [12] | 0 |
| 6.39 | Gibson [27] | 0 | 8.47 | Mandy [92] | 0 |
| 6.18 | Botting [87] | 0 | 8.14 | Bambini [60] | 1 |

(**A**) Scopus

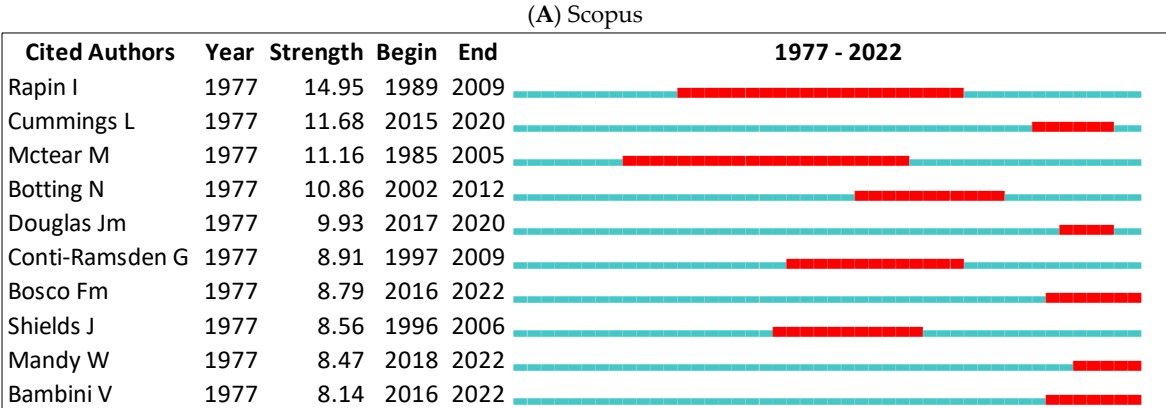

**Figure 13.** *Cont.*

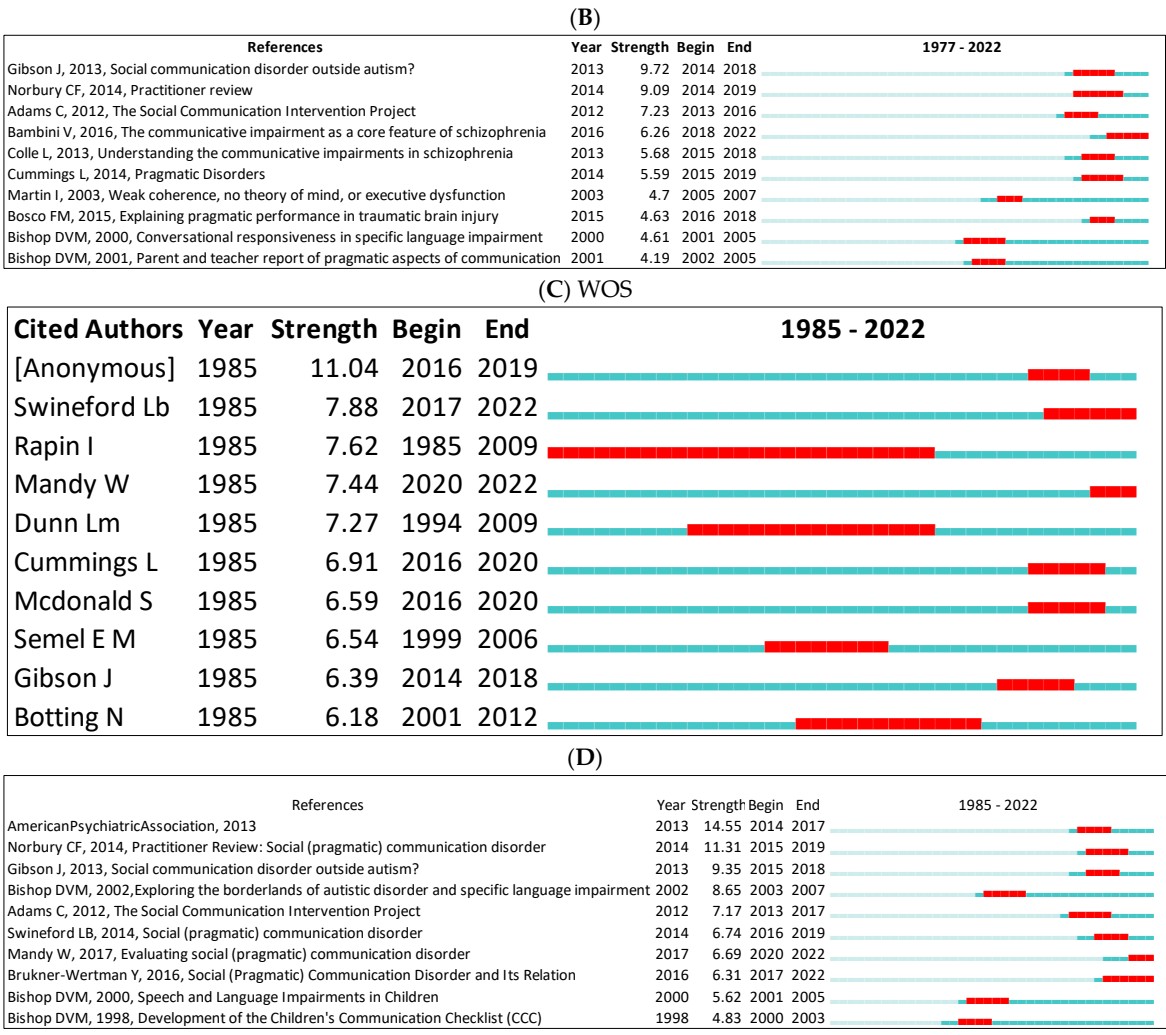

**Figure 13.** Top 10 Cited Authors and References with the Strongest Citation Bursts.

### 3.4.4. Centrality

In Scopus, the top-ranked item by centrality is Bishop [20] in Cluster #0, with a centrality of 109. The second one is Martin [100] in Cluster #1, with centrality of 99. In the WOS, the top-ranked item by centrality is Adams [81] n Cluster #0, with centrality of 134. The second one is Bishop [81] in Cluster #0, with a centrality of 127. Details are provided in Table 7.

**Table 7.** Centrality for Pragmatic Language Impairment Using Scientometric Analysis.

| | WoS | | | Scopus | |
|---|---|---|---|---|---|
| **Centrality** | **Reference** | **Cluster ID** | **Centrality** | **Reference** | **Cluster ID** |
| 134 | Adams [81] | 0 | 109 | Bishop [20] | 0 |
| 127 | Bishop [81] | 0 | 99 | Martin [100] | 1 |
| 123 | American Psychiatric Association [6] | 0 | 99 | McDonald [101] | 1 |
| 111 | Baron Cohen [102] | 3 | 97 | Rapin [17] | 0 |
| 99 | Bishop [103] | 1 | 97 | Lord [80] | 2 |
| 92 | Rapin [7] | 0 | 94 | Adams [81] | 0 |
| 72 | Botting [87] | 0 | 92 | Baron-Cohen [72] | 4 |
| 70 | Conti-Ramsden [104] | 1 | 91 | Botting [87] | 0 |
| 67 | Brinton [105] | 5 | 84 | Norbury [77] | 0 |
| 65 | Rutter [84] | 0 | 81 | Conti-Ramsden [96] | 0 |

3.4.5. Sigma

In Scopus, the top-ranked item by sigma is Bishop [20] in Cluster #0, with a sigma of 0.00. The second one is Martin [100] in Cluster #1, with a sigma of 0.00. In WOS, the top-ranked item by sigma is Adams [81] in Cluster #0, with a sigma of 0.00. The second one is Bishop [81] in Cluster #0, with a sigma of 0.00. See Table 8 for more detail.

**Table 8.** Sigma Metrics for Pragmatic Language Impairment Using Scientometric Analysis.

| WoS | | | Scopus | | |
|---|---|---|---|---|---|
| Sigma | Reference | Cluster ID | Sigma | Reference | Cluster ID |
| 0 | Adams [81] | 0 | 0 | Bishop [20] | 0 |
| 0 | Bishop [81] | 0 | 0 | Martin [100] | 1 |
| 0 | American Psychiatric Association [6] | 0 | 0 | McDonald [101] | 1 |
| 0 | Baron Cohen [102] | 3 | 0 | Rapin [17] | 0 |
| 0 | Bishop [103] | 1 | 0 | Lord [80] | 2 |
| 0 | Rapin [7] | 0 | 0 | Adams [81] | 0 |
| 0 | Botting [87] | 0 | 0 | Baron-Cohen [72] | 4 |
| 0 | Conti-Ramsden [104] | 1 | 0 | Botting [87] | 0 |
| 0 | Brinton [105] | 5 | 0 | Norbury [77] | 0 |
| 0 | Rutter [84] | 0 | 0 | Conti-Ramsden [96] | 0 |

## 4. Discussion

In this scientometric review, we embarked on a systematic exploration of the progression and state of research on PLI, a condition that is characterised by difficulties in language use within social contexts, turn-taking, and contextual understanding. Our results were presented in two segments, the first dealing with bibliometric indicators such as publications by year, top-contributing countries, universities, journals, publishers, subject/research domains, and authors. The second segment delved into scientometric indicators, such as citation, co-citation, and co-occurrence indicators.

To answer the first question, the bibliometric indicators reveal seven important observations. First, research on PLI has seen a notable increase in the past two decades. This suggests a potential role for these technologies in the diagnosis and treatment of PLI. Second, the United States, the United Kingdom, and Australia emerged as the primary contributors to the body of PLI research. Third, universities in the UK in particular were found to be the most prolific institutions in the field. Fourth, the journals publishing most frequently in this area were those associated with speech and language disorders, autism, clinical linguistics, and neurolinguistics. Fifth, Elsevier and Wiley were the primary publishers. Sixth, the domains of medicine, psychology, social sciences, and linguistics were frequently connected to PLI research. The seventh and final observation revealed Adams [61], Bishop [9], and Cummings [59] as key contributors to the field.

To answer the second question, merging the bibliometric findings with the scientometric indicators allows us to draw four significant implications. First, the most frequently searched keywords in PLI research indicate the popular topics of interest within the field. These include language disability [106], social communication disorder [43], language disorder [107], autism spectrum disorder [108], verbal behaviour [109], conversational characteristics [79], pragmatic language impairment [110], deficit [111], specific language impairment [112], and mind [113]. This insight can help shape the future direction of PLI research by identifying the areas of highest interest or greatest concern. Using bibliometric data, we identified the top 10 cited articles in each database. These explored various topics assessment [9], autism [9,76], and gaze behaviour [63,64].

Second, our analysis of 2,852 documents related to PLI yielded several major co-citation clusters, including social communication disorder [43], traumatic brain injury [114], autism spectrum disorder [108], and inferential meaning [115]. Further exploration of

these clusters revealed four distinct patterns that could guide subsequent research and therapeutic strategies.

Third, by combining bibliometric and scientometric indicators, we could identify the most influential, central, and productive authors in the field. For example, Adams, Bishop, and Lockton have been instrumental in exploring the connections between PLI, autism, and developmental language disorder [61,116,117]. Cummings has made significant contributions to our understanding of pragmatic impairment following COVID-19 [59].

Fourth, while numerous authors have contributed to the advancement of PLI research, some have garnered more attention due to the quantity, quality, or topical relevance of their work. Using sigma metrics, we identified authors who might experience rapid growth in citation due to their focus on key topics, including referential communication [81], Asperger's syndrome [20], autism [118], and traumatic brain injury [100,101].

Building on the initial part of the discussion, the relationship between the rise of PLI and the interdisciplinary nature of PLI is remarkable. As highlighted in "The Rise of Pragmatic Language Impairment", the recognition of PLI as a distinct condition has grown considerably in the past two decades, concurrent with the substantial progress in interdisciplinary studies. Developments in interdisciplinary studies have not only provided new insights for diagnosing and treating PLI but have also broadened the horizons for research, facilitating more in-depth and nuanced studies.

The section "Pragmatic Language Impairment and Autism Spectrum Disorder" in our introduction hinted at the intertwined relationship between these two conditions. This connection was evidently mirrored in our findings. The frequent co-citation of these terms suggests a shared focus in the research community. It also reflects the ongoing debate about the overlap and distinctions between these conditions. This pattern reinforces the importance of continued investigation into the relationship between PLI and Autism Spectrum Disorder to enhance our understanding and refine diagnostic and therapeutic strategies.

Our review of "Diagnostic Instruments for Pragmatic Language Impairment" and "Scientific Contributions for Pragmatic Language Impairment" established a context for the key players and tools in the field. This context was reflected in our findings as some of the most influential authors have made significant contributions to developing diagnostic tools and furthering our understanding of PLI. The prominence of papers exploring assessment tools in our top-cited articles also indicates the importance of reliable and valid instruments in PLI research.

In conclusion, the themes outlined in our introduction provided a plausible background for our scientometric review, drawing a comprehensive picture of PLI research. The connections between these themes and our findings highlight the importance of understanding the historical, diagnostic, and scientific context to appreciate the current state and future directions of PLI research.

## 5. Conclusions

The importance of accurate interpretation of findings derived from scientometric studies cannot be overstated, particularly for researchers working in the field of PLI [119]. As the popularity of scientometric research methods has continued to grow in recent years [120,121], it has become crucial to ensure that the data collection and analysis approaches employed are both comprehensive and rigorous.

To enhance the robustness of scientometric studies, authors are highly recommended to gather data from multiple sources rather than relying solely on a single database, unless specific circumstances entirely justify such an approach. In our study, we took this into account and gathered data from three different databases, namely Scopus, WOS, and Lens, to provide a more comprehensive view of the research landscape in this area.

Moreover, incorporating various analysis tools in the research process allows for the inclusion of a diverse range of scientometric indicators, which can help to generate more detailed and informative insights. In our study, we utilised both CiteSpace and VOSviewer for our analysis, enabling us to examine different aspects of the research landscape and

identify key trends, influential publications, and emerging topics in the field of pragmatic language impairment.

By adopting such an approach, researchers can ensure that their scientometric studies are more comprehensive, reliable, and informative, ultimately contributing to a better understanding of the research dynamics in their respective fields. This, in turn, can help to identify knowledge gaps, inform future research directions, and support evidence-based decision-making within the scientific community.

Further, this study has two significant theoretical implications. Firstly, it underscores the need for dedicated research centres focusing on the investigation of PLI. Ideally, these centres should assemble a team of experts that understand the interdisciplinary nature of pragmatics and the complexity of PLI. These experts would range from clinicians, psychologists, and speech and language therapists to educators and researchers. A robust and comprehensive exploration and understanding of PLI should encompass all existing evidence—from its incidence and prevalence to its diagnosis, assessment, and treatment strategies. This should also extend to future prospects, including potential advancements in diagnostic tools, treatment modalities, and theoretical frameworks.

The second implication stems from the need for a comprehensive, accessible database that collates all theoretical, historical, and empirical evidence related to PLI. This database would serve as a central repository of knowledge, documenting the evolution of PLI research and facilitating more efficient and effective research outcomes in the future. By allowing researchers and practitioners easy access to a wealth of information on PLI, we can encourage greater collaboration, cross-pollination of ideas, and ultimately, more impactful research and interventions for individuals with PLI.

**Author Contributions:** Conceptualisation, S.A. and T.T.; Data curation, A.A.; Formal analysis, A.A.; Funding acquisition, H.A.; Investigation, H.A.; Methodology, A.A.; Project administration, A.A.; Resources, H.A., S.A. and T.T.; Software, A.A.; Supervision, H.A. and S.A.; Validation, A.A. and H.A.; Visualisation, A.A.; Writing—original draft, A.A. and T.T.; Writing—review & editing, A.A. and H.A. All authors have read and agreed to the published version of the manuscript.

**Funding:** The research was funded by King Saud University, Riyadh, Saudi Arabia, under the research project RSP2023/R251.

**Institutional Review Board Statement:** This research did not require IRB approval.

**Informed Consent Statement:** Neither human nor non-human subjects were involved directly in this research. Therefore, informed consent was not required.

**Data Availability Statement:** The data presented in this study are available on request from the first author.

**Acknowledgments:** The authors would like to acknowledge the financial support for publication from King Saud University, Riyadh, Saudi Arabia for funding the publication of this research (RSP2023/R251).

**Conflicts of Interest:** The authors declare no conflict of interest.

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
