# Peer review of "Pragmatic Language Impairment: A Scientometric Review"

_applsci, doi:10.3390/app13169308_

Round 1

Reviewer 1 Report (Previous Reviewer 1)

The authors have not taken into consideration any of the suggestions of this reviewer. Their argument does not justify why the suggested changes do not seem appropriate.

For example, to the suggestion to improve the format of Table 3 they indicate: "....

I believe that this may not be necessary as the focus is on the documents themselves, rather than their ranking. As for the structure of the table, we could either create a separate table for each database or use a more complex table that would be more practical.”

However, this reviewer believes that tables in a scientific article are a concise and effective way of presenting large amounts of data. For this purpose, care must be taken in the design, so that legends are clear and concise, data should be classified into categories, with sufficient spacing between columns and rows, and with a legible font type and size.

Regarding the format of the figures, they argue that what they want to highlight is: “The intensity of the text correlates with the amount of research conducted in that particular area related to PLI. Although the text itself may be illegible, the crucial factor is the intensity, which indicates a higher level of research in that specific theme.”

However, I believe that the figures should draw the reader's attention and communicate the most significant results visually and clearly.

Even considering that the visual elements (figures and tables) are used to communicate large amounts of complex information that would be complicated to explain in the text. In this work, the number of figures and tables in the results section is excessive (more than 30 figures) without presenting a text explanation of the data obtained.

Regarding the main practical implications of this work, the authors still do not respond either in the text of the paper or when asked by the reviewer.

Author Response

Reviewer 2 Report (Previous Reviewer 3)

The manuscript can now be accepted. Thank you for your revised version.

Author Response

Reviewer 3 Report (New Reviewer)

In the present manuscript, the authors explore three databases to collect the bibliometric and scientometric of published works related to Pragmatic Language Impairment. Among their findings, the authors show that there has been an increase of publications on the topic (mostly in England and the USA), that publications mostly cluster around autism, social communication disorder, and traumatic brain injury and are in the fields of medicine, linguistics and rehabilitation among others.

I think the authors’ original intent as well as chosen methodology may be of great interest to the readers. However, I have major concerns with the aims and  discussion and overall quality of presentation.

My main concern is with the aims of the manuscript. It seems to me that these would be descriptive, as the review outcomes are data on the number of publications, areas of publications, journals, editors, clustered keywords, etc. and not on the outcomes of scientific research. Gathering this information on the topic would be in itself a strong motivation to run such a review. However, the authors push an angle that does not come out from the manuscript’s result or discussion sections, namely that of the “(scientometric) implications for diagnosis and rehabilitation”. Apart from the fact that it is not quite clear what the authors mean by “scientometric implications”, since scientrometric is related to the data they gather, this is a much different RQ that would be answered by extracting at least partially some outcomes from the papers dealing with PLI, which is not what the authors do here. This all becomes even more apparent in the discussion, where the is in fact no discussion of any implication for diagnosis and rehabilitation (the words “diagnosis” and “rehabilitation” only appear once in the discussion sections after the aims are repeated). Claims about the manuscript’s goals such as “This approach not only sheds light on the development of PLI as a concept but also highlights the implications of digital technologies in advancing our understanding, diagnosis, and rehabilitation of this complex condition.” (lines 241-244) are not in any way mirrored in the data extraction and in the discussion. The discussion further mentions that results were interpreted in light of their implications for “digital technologies”, but the word “digital” itself only appears once in the introduction before reappearing in the discussion as something that “was discussed” (lines 603-605) and it is therefore very unclear what is even meant by this. Since this is a central topic for the special issue, the authors should consider really revisiting the priority of this topic in their introduction, results, and analysis.

The discussion itself is a summary of main findings (that would have been fine had the authors not claimed to have theoretical aims), and a short description of what the authors think should happen in the future based on their understanding of the topic more than the results of the review or background literature (“centres of experts working on PLI should be established”).

A second major point is with the introduction in general. The choice of which topics to treat in the introduction is, in my opinion, questionable, and penalises the continuity between the introduction and the results section. Most notably, the manuscript deals with impairment in pragmatics, but this is never described in full with examples of pragmatic impairment and the relevant literature (which is lacking), which would also provide some circularity with the literature that is then found in the discussion of results.

In short, while I think the authors worked hard and their results might have potential, I believe the introduction and the discussion sections need to be heavily modified following simpler and clearer aims, and a clearer centrality of the topic of the S.I., before this manuscript may be considered for publication.

Some comments:

Line 27 – “based on these findings” – the findings are not reported in the abstract.

Line 36 – the subsections of 1 are numbered 2.x instead of 1.x. As a side note, Immediately starting the introduction with a subsection is not easy on the reader. I think what is now 2.1 can be part of the larger section with no need for a separate title.

Line 46- 48 – Why did they believe that the “semantic pragmatic syndrome” definition was excluding disorders such as ASD? I feel like there is some information missing here. The fact that structural language abilities would have to be mostly intact does not necessarily rule out ASD. The authors should explain R&A and B&R’s definitions better.

Line 53 – why did they come to this conclusion?

Line 55 – “after more than a decade of trying to highlight the essential symptoms” – These essential symptoms as they have described in the literature by different authors should be clearly discussed.

Line 56 – IPL?

Line 59 – Disorder or Impairment?

Line 62 – his > its

Line 72 – I don’t see a shift in topic from the previous subsection and this one, at least for the first three paragraphs.

Line 81 – 83 – this is the first “definition” we get of pragmatic skills. It should be expanded further with references and examples.

Line 86 – this back and forth between definitions is rather confusing. The authors do mention at the beginning that there have been shifts in terminology – It would be better if, after having stated this, they always referred to the disorder as PLI.

Line 100 – This definition needs more references.

Line 103-105 – I don’t think this is something R&A say specifically, it’s statistical, demographic data. Also, when such data are provided, it is usually best to have the most recent reference one can find – a lot can change in terms of diagnosis in almost 40 years, which is how old that paper is.

Line 110-114 – “According to the definition by the American Psychiatric Association [23], one of the symptoms of people with autism is a lack of understanding of the context of conversation, stereotypical and repetitive communication. This is supported by numerous studies showing the overlap of symptoms in people with PLI and people with ASD” – these sentences seem to be saying the same thing. Autism is characterised by PLI. Please restate or provide examples.

Line 114 – the authors should really expand on their references. There is a whole body of literature on pragmatic impairments in ASD that is missing.

Line 115 – it is confirmed by many authors, B and C-R being only one of the studies.

Lines 116-117 – I don’t follow – why does it call into question whether children with PLI have ASD if only half of the group in B&C-R was within the definition of ASD?

Line 128 – I am struggling to understand this. Disorders can be comorbid without needing to be grouped together. Please clarify.

Line 138 - Why is 2.4 relevant if there is no discussion on the use of these instruments in the literature?

Line 142 – number for the reference

Line 180-209 – This does not read as relevant enough to be a standalone subsection. If it is a testimony of how little research is done on the subject, this should be part of the section discussing research on PLI.

Line 224-225 – unclear, please rephrase and explain what is meant by “debatable concept”.

Line 234 – “the extent of knowledge production” – unclear, please rephrase.

Line 235 – the future trajectories?

Line 230-235 – please clearly state the research questions.

Line 257 – the difference between elements and type indicators is unclear.

Table 1 –The middle column seems unnecessary and removing it would make the table more readable. I believe “bibliometric” should be parallel to “scientometric” and between likes.

Table 2 – consider inserting the main keywords in the text in 2.3

Line 294 – cantered > centred

Line 334 – insert number of documents before and after duplicates were removed for each database (or state how many were removed by each)

Figure 10 – is there no way of visualising the full keywords?

Figure 11 – should contain key of the colour coding in the figure label.

Table 3 – Reporting Author and Year would make it easier to navigate than title.

Line 603-605 – The word “digital” only appears in the introduction before it reappears in the discussion as something that “was discussed”.

Line 613 – remove references as you are referring to the authors and not to specific papers (as far as I understand)

Line 690 – is this in alphabetical order or order of appearance?

Round 2

Reviewer 1 Report (Previous Reviewer 1)

No comments

Author Response

Thanks so much for your feedback. 

Reviewer 3 Report (New Reviewer)

The authors have addressed the major concerns I had raised, and the manuscript is greatly improved.

Some minor adjustments need to be made throughout the text (including some that were already pointed out but were not addressed), and the discussion section and conclusions should be reorganised on the basis of the RQs the authors have identified.

Comments:

-The abstract is better in terms of the aims, but it should be edited down, particularly from line 29 onwards.

-Line 53 now 76 why did they come to this conclusion?

I think with a basic knowledge of semantics and pragmatics it could be understood that they are two different fields regardless of their interconnection.

Really, this is the theoretical explanation Bishop gives in her 1998 paper? Fascinating!

The authors should either remove the comment or provide further details.

-Line 81 – now 107 - this is the first “definition” we get of pragmatic skills. It should be expanded further with references and examples.

Thanks but I am afraid that I have to disagree to this point because the paper is not supposed to discuss what is pragmatics in details as the assumption is to demonstrate research trends in PLI with shedding some light on what is PLI including pragmatics.

Of pragmatics? No. Of pragmatic language impairment? Absolutely. Examples of PLI are scattered across the paper. I suggest the authors have a clear and short description of what is meant by “impaired social use of verbal and nonverbal communication”, with a few examples.

-Line 110

“but they all ruled out the possibility of using pragmatics in nonverbal communication”

Unlcear, please rephrase.

-Lines 126-128

“I think a reference remains valid as long as it is an opinion not a finding and this statement here is an opinion of the author regardless of the time it was published—it remains valid.”

One cannot have an opinion on how many children with a disorder are undiagnosed until school years – this is a demographic factor. If the authors cited in the manuscript report this as a personal opinion/feeling, then it should not be referenced. The authors should either remove this or find confirmation of this in more recent literature.

-Lines 139-141

This statement should be removed. It is not within the scope of the paper and it opens up a much broader discussion that cannot be dismissed like this.

-Table 1 –The middle column seems unnecessary and removing it would make the table more readable. I believe “bibliometric” should be parallel to “scientometric” and between likes.

Thanks, but they are not actually the same. The former is generated automatically from the databases while the latter needs to be generated using a software.

What? This answer seems irrelevant to my point. And in fact, you did fix bibliometric and scientometric not being parallel in the table.

My second point was that the middle column makes the table hard to read. I suggest separating the definition of each bibliometric and scientometric label from this table and have it wither in a separate table or in the text.

-Line 597-600 Your aims are now stated as 3 RQs, so not twofold. Please amend.

-Please openly address your research questions 1-3 in the discussion.

-I believe that what the authors refer to ad theoretical and practical implications should simply be their Conclusions. Given the nature of the research these are not incisive enough to be standalone subsections but rather should be final considerations.

Author Response

This manuscript is a resubmission of an earlier submission. The following is a list of the peer review reports and author responses from that submission.

Round 1

Reviewer 1 Report

I consider that in the study entitled “Pragmatic Language Impairment: Analysis of Mapping ‎‎Knowledge Domains” a very precise bibliometric work is carried out. However, However, I point out some considerations regarding the content

1.       Presentation of the work: the authors pay little attention to the format of tables and figures.

For example, table 2, which presents the keywords used in the search carried out in the 3 databases used, is repetitive and confusing, since it is assumed that they should be the same in the 3 databases. It is not like this?

Table 3 should improve its format, since the data appears out of order.

The figures do not follow the same format, not even those that refer to the same content, only varying the database used.

The data presented in Figure 10 are not observed.

2.       Regarding the content: in the introduction some aspects that are important in the delimitation of the disorder are addressed (Diagnostic Instruments) but that are not included exhaustively in the bibliometric study.

In the practical implications the authors point out: “As a way to accomplish this, one must retrieve data from various sources and avoid limiting data retrieval to a single database unless it is entirely justified (e.g., in this study we used Scopus, WOS, and Lens as our sources)”. More than an implication, I understand that it is a suggestion, since it is assumed that when researchers carry out a bibliographical review in a field, they usually use several databases.

The authors point out two theoretical implications that, from my point of view, are ideal, perhaps they should indicate how to put them into practice.

Check references, for example 7, 88, 98 and 110

Reviewer 2 Report

I agree with the authors that the problem of pragmatic language impairment ‎‎(PLI) is of particular relevance in our time. I also highly appreciate the desire of the authors to use modern methods of bibliometric and scientometric analysis to study this problem.

However, the presented manuscript raises many methodological questions and doubts.

First of all, I do not agree with the authors that “this study examined the past, present, and future directions of the concept of PLI” (lines 24-25). This is mentioned a little in the introduction, but as a result of a voluminous bibliometric and scientometric analysis, no clear conclusions about the development of the concept are formulated.

The next incomprehensible question is why the authors took 3 bases (Scopus, WoS, and Lens) for analysis at once. Unfortunately, I could not understand whether there were duplicate publications in the three samples or not. In any case, the use of 3 databases suggests further comparison of the results obtained, but the article has only a description. In my opinion, it would be enough to take 1 database for analysis, but consider the results in more detail.

Next, the manuscript contains many Figures that take up a lot of space, which is not always justified. For example, data on the number of articles by year, country, journal, could be presented in comparative tables. Also Figure 10 (A-D) is absolutely "unreadable".

So, in line with the topic of the Special Issue, I would suggest that the authors "narrow down" the purpose of the article and do a similar analysis for the concept of “Digital Technologies for Diagnosis and Rehabilitation of PLI”.

Reviewer 3 Report

The structure of the paper is missing from the Introduction. Purpose of the Present Study is introduced on page 5. I would consider to remove it to the end of Introduction (page 2). 

My main criticism is that the article is too descriptive and not very interesting. From my point of view, that means a fairly major revision. However, if you are looking for this type of article for the issue, then it only needs minor modifications.  

The results section makes for a long read. It is very descriptive and any analysis is left for a 2 page discussion, which is also fairly descriptive and makes no attempt to paint a bigger picture for the reader. Unfortunately, for me as a reader, this makes it not very interesting. I would urge the authors to rethink their very extensive findings in the light of the purposes of their study, and frame it within a more in-depth contextualisation.